

# Inter-model comparison of global hydroxyl radical (OH) distributions and their impact on atmospheric methane over the 2000-2016 period

Yuanhong Zhao[1], Marielle Saunois[1], Philippe Bousquet[1], Xin Lin[1*], Michaela I. Hegglin[2], Josep G. Canadell[3], Robert B. Jackson [4], Didier A. Hauglustaine[1], Sophie Szopa[1], Ann R. Stavert[5], Nathan Luke Abraham[6, 7], Alex T. Archibald[6, 7], Slimane Bekki[8], Makoto Deushi[9], Patrick Jöckel[10], Béatrice Josse[11], Douglas Kinnison[12], Ole Kirner[13], Virginie Marécal[11], Fiona M. O'Connor[14], David A. Plummer[15], Laura E. Revell[16, 17] Eugene Rozanov[16, 18], Andrea Stenke[16], Sarah Strode[19, 20], Simone Tilmes[21], Edward J. Dlugokencky[22], and Bo Zheng[1]

[1] Laboratoire des Sciences du Climat et de l'Environnement, LSCE-IPSL (CEA-CNRS-UVSQ), Université Paris-Saclay, 91191 Gif-sur-Yvette, France

[2] Department of Meteorology, University of Reading, Earley Gate, Reading RG6 6BB, United Kingdom

[3] Global Carbon Project, CSIRO Oceans and Atmosphere, Canberra, Australian Capital Territory 2601, Australia

[4] Earth System Science Department, Woods Institute for the Environment, and Precourt Institute for Energy, Stanford University, Stanford, CA 94305, USA

[5] CSIRO Oceans and Atmosphere, Aspendale, Victoria, 3195, Australia

[6] Department of Chemistry, University of Cambridge, CB2 1EW, UK

[7] NCAS-Climate, University of Cambridge, CB2 1EW, UK

[8] LATMOS, Université Pierre et Marie Curie, 4 Place Jussieu Tour 45, couloir 45-46, 3e étage Boite 102, 75252, Paris Cedex 05, France

[9] Meteorological Research Institute, 1-1 Nagamine, Tsukuba, Ibaraki, 305-0052, Japan

[10] Deutsches Zentrum für Luft- und Raumfahrt (DLR), Institut für Physik der Atmosphäre, Oberpfaffenhofen, Germany





[11]Centre National de Recherches Météorologiques, Université de Toulouse, Météo-France, CNRS, Toulouse, France

[12]Atmospheric Chemistry Observations and Modeling Laboratory, National Center for Atmospheric Research, 3090 Center Green Drive, Boulder, CO, 80301, USA

[13] Steinbuch Centre for Computing, Karlsruhe Institute of Technology, Karlsruhe, Germany

[14] Met Office Hadley Centre, Exeter, EX1 3PB, UK

[15] Climate Research Branch, Environment and Climate Change Canada, Montréal, Canada

[16] Institute for Atmospheric and Climate Science, ETH Zürich (ETHZ), Zürich, Switzerland

[17] School of Physical and Chemical Sciences, University of Canterbury, Christchurch, New Zealand

[18] Physikalisch-Meteorologisches Observatorium Davos World Radiation Centre, Dorfstrasse 33, 7260 Davos Dorf

[19] NASA Goddard Space Flight Center, Greenbelt, MD, USA

[20] Universities Space Research Association (USRA), GESTAR, Columbia, MD, USA

[21] National Center for Atmospheric Research, Boulder, CO, USA

[22] Global Monitoring Division, NOAA Earth System Research Laboratory, Boulder, CO, USA

[*] Now at: Climate and Space Sciences and Engineering, University of Michigan, Ann Arbor, MI 48109, USA

*Correspondence to*: Yuanhong Zhao (yuanhong.zhao@lsce.ipsl.fr)



## Abstract

The modeling study presented here aims to estimate how uncertainties in global hydroxyl radical (OH) distributions, variability, and trends may contribute to resolve discrepancies between simulated and observed methane ($CH_4$) changes since 2000. A multi-model ensemble of 14 OH fields were analysed and were aggregated into 64 scenarios to force the offline atmospheric chemistry transport model LMDz with a standard $CH_4$ emission scenario over the period 2000-2016. The multi-model simulated global volume-weighted tropospheric mean OH concentration([OH]) averaged over 2000-2010 ranges between $8.7 \times 10^5$ and $12.8 \times 10^5$ molec $cm^{-3}$. The inter-model differences in tropospheric OH burden and vertical distributions are mainly determined by the differences in the nitrogen oxide (NO) distributions, while the spatial discrepancies between OH fields are mostly due to differences in natural emissions and VOC chemistry. From 2000 to 2010, most simulated OH fields show an increase of $0.1$-$0.3 \times 10^5$ molec $cm^{-3}$ in the tropospheric mean [OH], with year-to-year variations much smaller than during the historical period 1960-2000. Once ingested into the LMDz model, these OH changes translated into a 5 to 15 ppbv reduction in $CH_4$ mixing ratio in 2010, which represent 7%-20% of the model simulated $CH_4$ increase due to surface emissions. Between 2010 and 2016, the ensemble of simulations showed that OH changes could lead to a $CH_4$ mixing ratio uncertainty of $> \pm 30$ ppbv. Over the full 2000-2016 time period, using a common state-of-the-art but non-optimized emission scenario, the impact of [OH] changes tested here can explain up to 54% of the gap between model simulations and observations. This result emphasizes the importance of better representing OH abundance and variations in $CH_4$ forward simulations and emission optimizations performed by atmospheric inversions.



# 1 Introduction

The hydroxyl radical (OH) is the main oxidizing agent in the troposphere (Levy, 1972). OH is produced

by the reaction of water vapor with excited oxygen atoms ($O^{1D}$), produced by ozone ($O_3$) photolysis ( $\lambda$

<340nm). In the troposphere, OH is rapidly removed by reactions with carbon monoxide (CO), methane

($CH_4$) and non-methane volatile organic compounds (NMVOCs) to generate hydroperoxyl radical ($HO_2$)

or organic peroxy radicals ($RO_2$), resulting in a short lifetime of a few seconds (Logan et al., 1981;

Lelieveld et al., 2004). $HO_2$ and $RO_2$ can further react with nitrogen oxide (NO) to regenerate OH (Crutzen,

1973; Zimmerman et al., 1987). At high latitudes, such a secondary production plays an important role,

because the OH primary production is limited by the supply of $O^{1D}$ and water vapor (Spivakovsky et al.,

2000). The abundance of OH reflects the combined effects of atmospheric composition (tropospheric $O_3$,

and NO, CO, $CH_4$, and NMVOCs) and of meteorological factors such as humidity, UV radiation, and

temperature.

Due to its short lifetime, global [OH] is difficult to estimate from direct measurements. Current

understanding on global [OH] has been obtained either from inversion of 1-1-1trichloroethane (methyl

chloroform, MCF) (Prinn et al., 2005; Bousquet et al., 2005; Montzka et al., 2011; Rigby et al., 2017;

Turner et al., 2017), or using atmospheric chemistry models (Naik et al., 2013; Voulgarakis et al., 2013,

Lelieveld et al., 2016). The former approach relies on the fact that OH is the main sink of MCF and on

the hypotheses that emissions and concentrations of MCF are well known and well measured, respectively.

The latter approach relies on chemistry-transport modeling with chemistry schemes of varying complexity.

The global mass-weighted tropospheric mean [OH] in the 2000s calculated by atmospheric chemistry

models was found to be about    $11.5 \times 10^5$ molec cm$^{-3}$, with an inter-model dispersion of ±15% (Naik et

al., 2013; Voulgarakis et al., 2013). Atmospheric chemistry models usually calculate higher [OH] over

the Northern hemisphere than the Southern hemisphere (N/S ratio>1) (Naik et al., 2013) whereas MCF

and $^{14}CO$ observations indicate a N/S ratio slightly smaller than 1 (Brenninkmeijer et al., 1992; Bousquet



et al., 2005; Patra et al., 2014).


OH determines the lifetime of most pollutants and non-$CO_2$ greenhouse gases including $CH_4$, the second most important anthropogenic greenhouse gas after carbon dioxide ($CO_2$) (Ciais et al., 2013). About 90% of tropospheric $CH_4$ is removed by reacting with OH (Ehhalt et al., 1974; Kirschke et al., 2013; Saunois et al., 2016). The tropospheric $CH_4$ chemical lifetime against OH oxidation (global tropospheric $CH_4$

burden divided by annual $CH_4$ tropospheric loss by OH) calculated by the models that participated in the Atmospheric Chemistry and Climate Model Inter-comparison Project (ACCMIP) is $9.3\pm1.6$ years, and the $CH_4$ total lifetime including all sink processes is $8.3\pm0.8$ years (Naik et al., 2013; Voulgarakis et al., 2013), smaller than that of $9.1\pm0.9$ years lifetime constrained by observations (Prather et al., 2012).

The tropospheric $CH_4$ burden has more than doubled compared to the pre-industrial era due to anthropogenic activities and climates change, resulting in about $0.62$ W m$^{-2}$ additional radiative forcing (Etminan et al., 2016). The global mean $CH_4$ growth rate decreased to near zero in the early 2000s but resumed increasing at ~5ppbv yr$^{-1}$ since 2006 and reached more than 10 ppbv yr$^{-1}$ in 2014 and in 2018 (Ed Dlugokencky, NOAA/ESRL, 2019). The growth rate of $CH_4$ is determined by the imbalance of its

sources, primarily from anthropogenic activities (agriculture, waste, fossil fuel production and usage, and biomass burning) but also from natural emissions (mainly wetland and other inland waters), and sinks (OH oxidation, other chemical reactions with chlorine and oxygen radicals, and soil uptake). The precise scenario of the stagnation and renewed $CH_4$ growths still remains unclear (e.g. Rigby et al 2017; Saunois et al., 2017; Nisbet et al., 2019).


Several studies have linked such $CH_4$ variations to inter-annual variations and trend of OH. Based on MCF inversions, McNorton et al. (2016) concluded that an increase in [OH] significantly contributed to the stable atmospheric $CH_4$ before 2007; Rigby et al. (2008) found that a decrease of $4\pm14\%$ in [OH]



could partly explain the $CH_4$ growth between 2006 and 2007; Bousquet et al. (2011) found a smaller

decrease in [OH] (<1% over the two years) and attributed the increase in $CH_4$ mostly to enhanced

emissions over tropical regions; Montzka et al. (2011) also calculated a small inter-annual variation of

$2.3\pm1.5\%$ in [OH] during 1998 to 2007. More recently, based on multi-species box model inversions,

Rigby et al. (2017) and Turner et al. (2017) inferred a decrease of $8\pm11\%$ and 7% in [OH] during 2004-

2014 and 2003-2016 respectively. Both of these studies suggested that such a decrease in [OH] is

equivalent to an increase of more than 20 Tg $yr^{-1}$ in $CH_4$ emissions, and therefore could significantly

contribute to explain the post-2007 $CH_4$ atmospheric growth, although a solution with constant OH cannot

be discarded. Meanwhile, atmospheric chemistry models have calculated only a small increase of [OH]

(decrease in $CH_4$ lifetime) during the early 2000s, and usually show much smaller year-to-year variations

and long-term trends than most of the MCF-based inversions (e.g. Voulgarakis et al., 2013; Holmes et al.,

2013; Dalsøren et al., 2016). The discrepancy between individual process-based models and MCF-proxy

approaches, and the uncertainties, limit our ability to be conclusive on the role of [OH] changes to explain

the $CH_4$ changes over the past decades.

To better understand OH distributions, trends, and influences on $CH_4$ since 2000, we have performed an

inter-model comparison of 14 OH fields, including 11 derived from chemistry transport and chemistry-

climate models that took part in the phase 1 of the Chemistry-Climate Model Initiative (CCMI) (Hegglin

and Lamarque, 2015; Morgenstern et al., 2017), 2 from different configurations of the LSCE atmospheric

chemistry transport model LMDz-INCA (Hauglustaine et al., 2004; Szopa et al., 2013), and 1 from the

TRANSCOM 2011 inter-comparison exercise (Patra et al., 2011). Using this ensemble of OH fields, our

aim is to estimate a range for the contribution of changes in [OH] to the atmospheric $CH_4$ variations since

2000, and to relate this contribution to characteristics of the different OH fields. Compared with previous

inter-model studies of OH based ACCMIP model which simulated time-slices (constant emissions) (Naik

et al., 2013; Voulgarakis et al., 2013), OH fields from CCMI and INCA model simulations are time-



dependent, which allow us to address the question of year to year variability. In the following, our analysis

first provides a brief description of the OH fields used in this study and the LMDz offline model (section

2). Section 3 compares the OH fields, analyses the factors contributing to inter-model differences and

presents their inter-annual variability. Section 4 presents and discusses the impact of the different OH

fields on the global $CH_4$ burden and growth rates simulated by LMDz. Section 5 summarizes the results

and conclusion.


## 2 Method

### 2.1 OH fields

The CCMI project aims to conduct a detailed evaluation of atmospheric chemistry models in order to

assess uncertainties in the models' projections of various climate-related topics such as tropospheric

composition (Hegglin and Lamarque, 2015; Morgenstern et al., 2017). The CCMI OH fields used in our

study are obtained from 10 different models and 2 CCMI reference experiments: REF-C1 (covering the

time period 1960-2010) and REF-C2 (covering 1960-2100). The REF-C1SD experiment is not analyzed

since it is conducted by only a part of the models and covers only a shorter time period (1980-2010). The

REF-C1 experiment is driven by state-of-the-art historical forcings and sea surface temperatures (sst) and

sea ice concentrations (sic) based on observations, while the REF-C2 experiment is using either coupled

ocean and sea ice modules or prescribes sst and sic obtained from another climate model. Since the REF-

C1 experiment is supposed to be more realistic regarding sea surface conditions, our analysis focused on

OH fields from the REF-C1 experiment before 2010 and only tested the influences of OH on $CH_4$

simulations after 2010 by applying the inter-annual variability from the REF-C2 experiment. Detailed

descriptions of CCMI simulations can be found in Morgenstern et al. (2017).

In this study, we used only the CCMI models that include coupled tropospheric ozone chemistry as listed

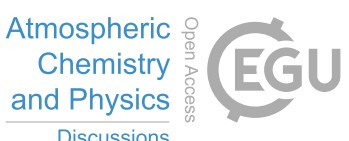

in Table 1. Note that EMAC offers fields at two different model resolutions. The level of detail in chemical

mechanism, in particular with respect to included NMVOCs, varies among the models. For example,

CMAM does not include any NMVOC species, but added 250 Tg CO emissions to account for CO

production from isoprene oxidation. UMUKCA-UCAM only include HCHO and SOCOL3 only include

HCHO and $C_5H_8$. Other models include multiple primary NMVOC species and more complex VOC

chemistry.


The anthropogenic emissions recommended for the two CCMI reference simulations are the MACCity

inventory (Granier et al., 2011) for 1960-2000. After 2000, the REF-C1 experiment continued to use the

MACCity inventory, while the REF-C2 used the RCP6.0 inventory (Masui et al., 2011). The CMAM

model did not follow this procedure and used the ACCMIP historical database of emissions (Lamarque

et al., 2010) until 2000 followed by RCP8.5 emissions (Riahi et al., 2011) instead. CCMI model

simulations also include natural emissions from lightning, soil and biogenic sources. Lightning $NO_x$

emission are calculated based on meteorological     data such as cloud top height (Price and Rind, 1994;

Grewe et al., 2001) and updraft mass flux (Allen and Pickering, 2002). Soil $NO_x$ emissions are calculated

interactively in EMAC and GEOSCCM using the scheme described by Yienger and Levy (1995), but are

prescribed in other models. Biogenic NMVOC emissions in CESM and GEOSCCM are calculated based

on the distribution of plant functional types and meteorology conditions with MEGAN, whereas the other

models apply prescribed biogenic NMVOC emissions.

The CCMI models do not represent $CH_4$ emissions explicitly but prescribe $CH_4$ surface mixing ratios

according to the RCP6.0 scenario (global mean of ~1750ppbv averaged from over 2000-2010) with

different spatial distributions: GEOSCCM, CESM and EMAC models consider the full latitudinal

gradient and prescribe $CH_4$ surface mixing ratios about 50 ppbv higher over the Northern hemisphere

than over the Southern hemisphere; while CMAM, MRI-ESM1r1, and SOCOL3 use global uniform



values. Photolysis rates are calculated either following online schemes such as FAST-JX (Neu et al., 2007;

Telford et al., 2013) by GEOSCCM, HadGEM3-EA, UMUKA-UCAM, JVAL (Sander et al., 2014) by

EMAC, or based on look-up tables with online cloud corrections by the rest of the models used in this

study. Kinetics and photolysis data are mainly from Sander et al. (2011) with a few exceptions. More

information on model characteristics can be found in Morgenstern et al. (2017) and references listed in

Table 1.


Additionally to CCMI OH fields, we also included 2 OH fields simulated by the Interaction with

Chemistry and Aerosols (INCA) coupled to the general circulation model of the Laboratoire de

Meteorologie Dynamique (LMD), LMDz (Sadourny and Laval, 1984; Hourdin and Armengaud, 1999;

Hourdin et al., 2006; Hauglustaine et al., 2004). The two INCA simulations are driven by different

versions of the LMDz GCM (INCA NMHC-AER-S covering time period 2000-2010 (Terrenoire et al.,

2019), and INCA NMHC covering time period 2000-2009 (Szopa et al., 2013)), which provide different

water vapor fields, and include different chemistry and emissions. The INCA NMHC-AER-S used the

latest version of INCA model including both gas phase (NMHC) and aerosol (AER) chemistry in the

troposphere and the stratosphere (S) (Terrenoire et al., 2019), while INCA NMHC used a former version

that only includes tropospheric gas-phase chemistry (Szopa et al., 2013). Anthropogenic emissions for

INCA NMHC-AER-S and INCA NMHC are from the Evaluating the Climate and AirQuality Impacts of

Short-Lived Pollutants (ECLIPSE) inventory (Stohl et al.,2015) for 2005 and from the RCP85 emission

inventory (Riahi et al., 2011)) for 2010, respectively.

Finally, we included in this study the OH field used in TransCom simulations, which results from a

combination of the semi-empirical tropospheric 3-dimensional OH field from Spivakovsky et al. (2000)

and a 2-dimensional simulated stratospheric OH for year 2000. The tropospheric OH was calculated using

prescribed chemical species ($O_3$, nitrogen oxides, and CO) as well as meteorological fields (temperature,

humidity, and cloud optical depth) to fit the observations. The original tropospheric [OH] has been

reduced by 8% to match $CH_3CCl_3$ observations (Patra et al., 2011). The TransCom OH field is only

climatological (one year of monthly fields).

In total, we compared 14 OH fields: 11 from CCMI, 2 from the online LMDz-INCA model and 1 from

TransCom. We analyzed spatial distributions and annual variations of OH fields by calculating volume-

weighted tropospheric mean [OH]. Since employing different weightings can results in large differences

in mean [OH] (Lawrence et al., 2001), we also calculated dry air mass-weighted tropospheric mean [OH]

to better compare with previous studies.

## 2.2 LMDz model simulations

2.2.1 Model description and setup

We have run the offline version LMDz5B of the LMDz model (Locatelli et al., 2015) at a horizontal

resolution of $3.75\,°\times1.85\,°$ with 39 vertical layers up to 3 hPa to access the impact of OH on tropospheric

$CH_4$. All monthly mean OH fields have been interpolated to the LMDz model grid. The transport of

atmospheric tracers is driven by prescribed air mass fluxes provided by the general circulation model

LMDz with horizontal wind fields nudged to ERA-Interim re-analysis meteorology data produced by the

European Center for Medium-Range Weather Forecasts (Dee et al, 2011). The vertical transport is

parameterized according to updates of the Emanuel (1991) scheme for convection and of the Louis (1979)

scheme for boundary layer mixing (Hourdin et al., 2016; Locatelli et al., 2015). Chemical sinks of $CH_4$

are calculated using prescribed three-dimensional OH and $O^{1D}$ fields. No chlorine-related sink is

simulated in this version of the model. To assess the influences of OH only, all LMDz simulations used

the same $O^{1D}$ fields generated by INCA model simulations. The reaction rate co-efficient (k) for $CH_4$

destruction by OH in the model is computed depending on temperature following Sander et al. (2011):



$$k = 2.45 \times 10^{-12} e^{-1775 \times (\frac{1}{T})} \quad\quad (1)$$

The LMDz model has been applied in various studies focusing on long-lived gases such as $CH_4$, $CO_2$ and

MCF (Bousquet et al., 2005; Pison et al., 2009; Lin et al., 2018). It has also been used in model inter-

comparison projects such as the TransCom experiment (Patra et al., 2011) with the simplified chemistry

module SACS (Pison et al., 2009) and CCMI (Morgenstern et al., 2017) but only with the stratospheric

chemistry model PEPROBUS (Jourdain et al., 2008).

The $CH_4$ emissions input to LMDz simulations are provided by the Global Carbon Project (GCP) methane

and include anthropogenic and biofuel emissions from EDGARv4.3.2 (Janssens-Maenhout et al., 2017),

the mean wetland emissions from Poulter et al., (2017), fire emissions from the Global Fire Emissions

Database Version 4.1 (GFED4) (Randerson et al., 2018), termite emissions as described by Saunois et al.

(2016), geological emissions based on the spatial distribution of Etiope (2015), ocean emissions from

Lambert and Schmidt (1993) and soil uptake from Ridgwell et al. (1999). EDGARv4.3.2 data, available

until 2012, were extrapolated from 2013 to 2016 using economical statistics according to the methodology

described by Saunois et al., (2016). Anthropogenic and fire emissions vary from 2000 to 2016 while

natural emissions are applied as a climatology.

The spatial distributions and annual variations of the $CH_4$ emissions during the study period are shown in

Fig.1. $CH_4$ emissions range from 10 to 40 kg ha$^{-1}$ yr$^{1}$ over most natural ecosystems and can exceed 100

kg ha$^{-1}$ yr$^{-1}$ over wetlands in Canada, South America, and Central Africa, as well as over densely

populated regions such as South and East Asia. Global net $CH_4$ emissions (soil uptake included) increased

by 15% from 482 Tg yr$^{-1}$ in 2000 to 552 Tg yr$^{-1}$ in 2016. Of this 70 Tg yr$^{-1}$ increase, 60 Tg yr$^{-1}$ (85%) are

emissions from the Northern hemisphere, mainly contributed by livestock (18 Tg yr$^{-1}$, 25%), oil and gas

(16 Tg yr$^{-1}$, 23%), coal burning (17 Tg yr$^{-1}$, 24%) and waste (13Tg yr$^{-1}$, 18%). The three emission peaks

in 2002, 2006 and 2015 are driven by biomass burning. This CH$_4$ emission scenario is state-of-the-art but has not been optimized for the simulated CH$_4$ mixing ratios to fit the observations.

2.2.2 Model simulations

Two sets of experiments (steady-state and transient simulations) have been performed to examine the impacts of the input OH fields on the global CH$_4$ burden as well as the CH$_4$ spatial distribution and annual variation. These tests excluded the OH fields from CESM1-CAM4chem and EMAC-L47MA, since they are similar to those of CESM1-WACCM and EMAC-L90MA, respectively. We also discarded the OH

fields from HadGEM3-ES and UMUKCA-UCAM because output from these two models has been supplied on too coarse vertical pressure levels. Finally, 10 different OH fields (seven from CCMI, two from LMDz-INCA and one from Transcom) were used in the two sets of simulations.

Initially, for each OH field described in Sect. 3, we ran 30 consecutive years of LMDz simulations (with

recycled same emissions, sinks, and meteorology of the year 2000) to allow the simulation to reach a steady-state (CH$_4$ has an approximate lifetime of 9 years in the atmosphere). This step aims to examine the impact of the magnitude and distribution of OH on the global CH$_4$ burden.

Secondly, we performed transient simulations starting from the year 2000, which are forced by time-

varying OH fields as well as time-varying emissions and meteorology fields. In order to compare the impacts of the different OH fields on realistic CH$_4$ mixing ratios, for each simulation (except the one using the OH fields from INCA NMHC), the OH field has been scaled to get the same LMDz simulated CH$_4$ loss as the one calculated by INCA NMHC in 2000, as INCA is the OH field consistently obtained using the LMDz transport. Then a series of LMDz model simulations is conducted to investigate the

impact of the various OH fields on CH$_4$ growth rates between 2000 and 2016 as summarized in Table 2.



The standard simulations (Run_standard in Table 2) using the 10 different OH fields (7 are from CCMI REF-C1), included annual variations and were performed from 2000 to 2010. Since REF-C1 experiments are only available up to 2010, the influence of OH on $CH_4$ mixing ratios after 2010 have been tested based

on alternative scenarios. First, for CCMI simulations, we tested a scenario that takes into account the annual variability from the REF-C2 experiments (Run_REF-C2 in Table 2). Previous ACCMIP model experiments showed slightly decreasing or increasing [OH] from 2000 to 2030 according to the largest or lowest radiative forcing pathways (RCP8.5 or RCP2.6), respectively (Voulgarakis et al., 2013). Top-down approaches suggested that global OH decreased by 0.5-1% annually from 2003 to 2016 (Rigby et al.,

2017; Turner et al., 2017). In order to assess the recent change in OH, we tested two additional scenarios between 2010 and 2016: one with [OH] increase of +1‰ $yr^{-1}$ (Run_OH_inc) and one with [OH] decrease of -1% $yr^{-1}$ (Run_OH_dec) . To assess influences from OH alone, we also conducted additional simulations of the period 2000 to 2016 with annually repeated prescribed [OH] equal to the year 2000 (Run_fix_OH) for each OH field. The differences between these constant OH simulations and the

corresponding time-varying OH simulations indicate the impact of OH inter-annual variations and trends on atmospheric $CH_4$ changes.

## 3 Analysis of OH fields

### 3.1 Spatial distributions of tropospheric OH

Fig. 2 shows the spatial distributions of volume-weighted tropospheric mean [OH] averaged from 2000 to 2010. Based on the 14 OH fields we have assembled, the global mean volume-weighted tropospheric [OH] vary from $8.7 \times 10^5$ to $12.8 \times 10^5$ molec $cm^{-3}$. SOCOL simulated the highest [OH], which overestimation of [OH] have been reported by Staehelin et al. (2017). To better compare with previous

studies, we also calculated dry air mass-weighted tropospheric mean [OH] in table 4, which vary from $9.4 \times 10^5$ to $14.4 \times 10^5$ molec $cm^{-3}$. This (large) range is consistent with previous multi-model results given by IPCC (2011) and the ACCMIP project (Naik et al., 2013; Voulgarakis et al), as well as with inversions



based on MCF observations (Bousquet et al., 2005; Rigby et al., 2017). The model spread remains large

as ~50% of the minimum value, as noted in previous studies (e.g. Naik et al., 2013).


Table 3 summarizes their inter-hemispheric ratios of tropospheric OH and mean values over four

latitudinal bands. The inter-hemispheric ratios (N/S ratios) of CCMI and INCA OH fields are within the

range of 1.2-1.5, similar to those from the ACCMIP project (Naik et al., 2013). In contrast, the TransCom

OH field has a ratio of 1.0, which is more consistent with that of MCF and $^{14}$C constrained OH

(Brenninkmeijer et al., 1992; Krol and Lelieveld, 2003; Bousquet et al., 2005). However, as discussed by

Spivakovsky et al. (2000), the TransCom OH field may overestimate Southern Extra-tropics OH by ~25%.

The lower N/S ratios inferred from MCF observations are mainly due to high [OH] over the Southern

Tropics (35% higher than Northern Tropics) (Bousquet et al., 2005). In contrast, process-based simulated

OH is 10-26% more abundant over the Northern Tropics than over the Southern Tropics, and 35% to >

90% higher over 30°N-90°N than 30°S-90°S in the CCMI models. Previous studies have attributed the

inconsistency between the simulated and the observed OH N/S ratios to a model overestimation of $O_3$ and

underestimation of CO over the Northern Hemisphere (Naik et al., 2013; Young et al., 2013; Strode et al.,

2015), as well as to a lack of OH recycling due to the presence of VOCs over rainforest (mainly located

in the Southern Tropics) (Lelieveld et al., 2008; Archibald et al., 2011).


We further assessed the simulated OH spread by comparing the detailed spatial distributions of OH fields

in Fig. 2. Nearly all CCMI models and two versions of the INCA model simulated high [OH] over eastern

North American and South and East Asia, which is related to higher tropospheric $O_3$ concentrations

(Cooper et al., 2014; Lu et al., 2018) and $NO_x$ emissions from human activities (Lamarque et al., 2010;

Miyazaki et al., 2012). High [OH] over these emission hotspots dominate the aforementioned simulated

large N/S ratio. Some models also simulated high OH values over the African savanna plains (MOCAGE

and INCA excluded), regions with intense biomass burning (van der Werf et al., 2006) and soil $NO_x$





emissions (Yienger and Levy 1995; Vinken et al., 2014). The $O_3$ concentrations used to generate the
TransCom OH field were larger in the Southern Tropics than in the Northern latitudes (Spivakovsky et

al., 2000), in contrast to recent observations (Cooper et al., 2014). Therefore, TransCom shows the highest
[OH] over the Southern Tropics during biomass burning seasons (Spivakovsky et al., 2000) and thus a
lower N/S ratio.

Despite consistency on high OH values over regions influenced by human activities and biomass burning,

models show the largest discrepancies over some natural ecosystem such as tropical rainforests. For
example, INCA, CESM, HadGEM3-Es, MRI-ESM1r1, MOCAGE and GEOSCCM simulated overall low
[OH] ($4 \times 10^5$-$14 \times 10^5$ molec cm$^{-3}$) over tropical rainforests, despite differences in details, while EMAC,
CMAM, SOCOL3 and UMUKCA-UCAM simulated overall high [OH] ($16 \times 10^5$- more than $25 \times 10^5$
molec cm$^{-3}$). Besides these, inter-model differences also exist in remote areas such as over the open ocean.

Most simulated OH fields show higher concentrations over continents or coastal areas due to higher
precursor emissions, while MRI-ESM1r1, EMAC, and GEOSGCM also simulated high values ($>15 \times 10^5$
molec cm$^{-3}$) over the open ocean. Factors contributing to these inter-model differences are further
discussed in Sect. 3.3

### 3.2 Vertical distributions


Figure 3 shows the vertical distribution of OH fields and Table 4 provides the volume-weighted mean
[OH] averaged over the troposphere and over three pressure latitudinal intervals representing the planetary
boundary layer, the mid-troposphere, and the upper troposphere (surface-750, 750-500, and 500-250 hPa,
respectively). At the global scale, the mean tropospheric concentration of TransCom OH increases by a

factor of nearly two from the surface ($7 \times 10^5$ molec cm$^{-3}$) to 600hPa ($13 \times 10^5$ molec cm$^{-3}$) and then
decreases rapidly with altitude ($7 \times 10^5$ molec cm$^{-3}$ at 250hPa). UMUKA-UCAM, HadGEM3-ES, CMAM,
MOCAGE, and SOCLO3 on the other hand all show a continuous decrease of [OH] with altitude from



the surface to the upper troposphere (e.g. the global mean concentrations of MOCAGE OH decreases

from $23.6 \times 10^5$ molec cm$^{-3}$ at the surface to $6.4 \times 10^5$ molec cm$^{-3}$ at 250hPa). Other OH profiles show much

smaller vertical variations in the troposphere (standard deviations of mean value below 200hPa $< 2 \times 10^5$

molec cm$^{-3}$).

Model simulated OH vertical distributions can also be different over land versus ocean (Fig. 3) and

between the different latitudinal bands (Fig. S1). For example, SOCOL3 [OH] continuously decreases

with altitude over both, land and ocean; MOCAGE OH increases from the surface ($14.9 \times 10^5$ molec cm$^{-3}$) to 800hPa ($18.2 \times 10^5$ molec cm$^{-3}$) and then decreases over land but almost continuously decreases over

the ocean; CMAM and UMUKCA-UCAM only show significant vertical variations in [OH] over land.

Vertical variations of most OH fields can be attributed to mid and low latitude regions, except for those

of SOCOL3 and MOCAGE, that also decrease with altitude over mid and high northern latitudes (45 °N

-90 °N, see Fig. S1).

### 3.3 Factors contributing to inter-model differences

Tropospheric OH is produced primarily through the reaction of O$^{1D}$ with H$_2$O and secondarily through

the reaction of NO with HO$_2$ and RO$_2$, and is removed primarily by reacting with CO and CH$_4$ (Logan et

al., 1981). Hence, factors controlling inter-model OH discrepancies can be complex as differences in

model emissions, chemistry, and dynamics can together impact [OH]. Here we propose a qualitative

analysis focusing on both, emissions and chemical mechanisms. A more quantitative analysis would

require a detailed model output of production and loss pathways and is beyond the scope of this work.

To analyse inter-model differences in OH vertical distributions, we compared the O$^{1D}$ photolysis rates,

specific humidity, and CO and NO mixing ratios in table 5. The inter-model variations (calculated as

standard deviation/multi-model mean) in tropospheric O$^{1D}$ photolysis rates, specific humidity, and CO



mixing ratios are usually <10%-20%, while NO mixing ratios show a larger variation of 38% (12-32pptv).

MRI-ESM1r1 simulated the highest NO tropospheric mixing ratio, mainly attributable to high values

above 200hPa, where OH formation is limited by $H_2O$. In addition, MRI-ESM1r1 has ~20% more CO

emissions than MOCAGE and GEOSCCM (Fig. S2), leading to about 10 ppbv higher CO mixing ratios,

offsetting (for [OH]) its higher $NO_x$ emissions and NO mixing ratios. The high NO mixing ratios near the

surface and mid-troposphere simulated by SOCOL3 (48 pptv below 750hPa and 10 pptv from 750 to

500hPa), MOCAGE (26 pptv below 750hPa and 14 pptv from 750 to 500hPa) and CMAM (17 ppbv

below 750hPa) are consistent with their high tropospheric and near-surface [OH].

Besides emissions, previous studies have reported additional factors leading to high surface NO and $NO_2$.

The overestimation of NO by MOCAGE could be due to the lack of $N_2O_5$ heterogeneous hydrolysis on

tropospheric aerosol, which is an efficient sink for $NO_x$ (Teyssèdre et al., 2007). SOCOL3 does not include

$N_2O_5$ heterogeneous hydrolysis and also overestimates tropospheric NO production by $NO_2$ photolysis

compared to other models, due to issues with the look-up tables used in the calculation of photolysis rates

(Revell et al. 2018). We conclude here that physical and chemical processes related to NO production and

loss can have a large impact on OH burden and its vertical distribution. In this context, an improved

representation of the partitioning between NO and other nitrogen species in the models seems of great

importance to correctly simulate tropospheric [OH].

Concerning the spatial distributions, as aforementioned in Sect. 3.1, the largest model discrepancies are

found over tropical rain forests. The [OH] over tropical rainforest regions are mostly sensitive to natural

emissions including $NO_x$ and NMVOCs, which vary among the models. Previous studies showed that

[OH] is more sensitive to soil and lightning emissions than to wildfires, because the former sources only

emit $NO_x$ (OH source), whereas the latter emits $NO_x$, CO and VOCs together (OH sources and OH sinks,

see Murray et al., 2014). Soil $NO_x$ emissions in CCMI models range from around 4 Tg N yr$^{-1}$ in MOCAGE





to more than 7 Tg N yr$^{-1}$ in GEOSCCM and 9Tg N yr$^{-1}$ in CMAM (Naik et al., 2013; Yienger and Levy, 1995). In particular, lower NO$_x$ emissions over South America and Africa in MOCAGE might be linked to lower [OH] over this region (Fig. S2). Isoprene and other NMVOCs remove about 3% and 7% of tropospheric OH on a global scale, respectively (Spivakovsky et al., 2000; Murray et al., 2014) and can be more important over tropical regions with higher emission rates (Sindelarova et al., 2014). The higher [OH] over tropical rainforests simulated by CMAM and UMUKCA-UCAM may be due to a lack of or the lower OH destruction by VOCs in these models. Therefore, the inter-model differences in OH spatial distributions over tropical rainforests may result from differences in natural emissions of VOC species and different related chemical reactions.

## 3.4 Inter-annual variations of OH

Figure 4 shows the time series of volume-weighted tropospheric mean [OH] from 1960 to 2010 (from REF-C1 CCMI comparison). During this period, all OH fields show small year-to-year variations, remaining within ±0.5×10$^5$ molec cm$^{-3}$. CCMI models simulated significantly different OH long-term evolutions from 1960 to 1980. For example, [OH] continuously decrease in the CMAM and HadGEM3-ES simulations (~-0.3×10$^5$ molec cm$^{-3}$); and increase in SOCOL3 (~+0.6×10$^5$ molec cm$^{-3}$), UMUKCA-UCAM (~+0.5×10$^5$ molec cm$^{-3}$), and MOCAGE (~+0.5×10$^5$ molec cm$^{-3}$) during this period, while other models show no obvious long-term trend. After 1980 (1990 for CMAM), all models show stabilized or slightly increasing [OH]. For our period of interest (after 2000) and focusing on the anomaly in [OH] compared to the 2000-2010 mean (Fig. 4b), OH year-to-year variations are found to be smaller than in previous decades and [OH] only increases by about 0.1-0.3×10$^5$ molec cm$^{-3}$ from 2000 to 2010.

Previous atmospheric chemistry model studies have concluded that anthropogenic activities lead to only a small perturbation of the OH burden, as the increased OH production tend to be compensated by an increased loss through reactions with CO and CH$_4$ (Lelieveld et al., 2000; Naik et al., 2013). The ensemble



of ACCMIP models simulated large divergent OH changes (even in their signs) from 1850 to 2000, but revealed a consistent increase of 3.5±2.2% from 1980 to 2000 (Naik et al., 2013). Here, for the same

period the increase of CCMI [OH] is 4.6±2.4%, consistent with the ACCMIP project (Naik et al., 2013) and with other atmospheric chemistry model studies (Dentener et al., 2003; John et al., 2012; Holmes et al., 2013; Dalsøren et al., 2016). The year-to-year variations of most CCMI and INCA OH fields are much smaller than the OH inter-annual variability based on MCF observations (e.g. Bousquet et al., 2005; Montzka et al., 2011), which can reach 8.5±1.0% from 1980 to 2000, and 2.3±1.5% from 1998 to 2007,

as compared to 2.1±0.8% and 1.0±0.5% here for these two periods.

We further analyzed regional [OH] trends from 2000 to 2010 in Fig. 5. Instead of dividing subdomains as Naik et al. (2013) did, we calculated the trend for each model grid-cell to identify and distinguish regions with different trends. Most models show significant positive [OH] trends over tropical regions (0.05-

$0.1 \times 10^5$ molec cm$^{-3}$ yr$^{-1}$) and over East and Southeast Asia (>$0.1 \times 10^5$ molec cm$^{-3}$ yr$^{-1}$). From 2000 to 2010, $NO_x$ emissions in the MACCity (RCP85) inventory increased by 83% over East Asia, which is much faster than the CO increase (8%) (Riahi et al., 2011) and can explain the strong positive trend of CCMI simulations. Over the rest of the extra-tropical regions such as North America and Western Europe, the models disagree on the sign of OH change. In the Southern hemisphere, where biogenic and fire emissions

dominate, most OH fields do not show clear trends and the inter-model differences are even larger. For example, MOCAGE simulated and OH decrease of >$0.1 \times 10^5$ molec cm$^{-3}$ yr$^{-1}$ over the Amazon, South Africa, and Indonesia, whereas MRI-ESM and EMAC-L90MA simulated positive OH trends over these regions; HadGEM3-ES simulated a significant decrease of the OH trend over most of the extra-tropical regions in the Southern hemisphere, while CMAM simulated slightly positive trends in this hemisphere.


In the following, we investigate how the differences in mean [OH] and variations presented in this section affect $CH_4$ burden and its variations for the period 2000-2016.





## 4 Influences of OH fields on CH$_4$ simulations


### 4.1 Global total CH$_4$ burden

We now present the results based on the first set of LMDz experiments, where the LMDz model was run for 30 years recycling the year 2000 until the steady-state is reached. The simulations using the OH fields as given by CCMI and INCA models provide a wide range of values for the tropospheric global mean CH$_4$ mixing ratios (Table 6), from 1204 ppbv (SOCOL3, with a global volume-weighted tropospheric mean [OH] of $12.8 \times 10^5$ molec cm$^{-3}$) to 1822 ppbv (INCA NMHC-AER-S, with a global volume-weighted tropospheric mean [OH] of $8.7 \times 10^5$ molec cm$^{-3}$). It appears that the global CH$_4$ burden is not only sensitive to the global mean [OH], but also to its vertical distribution. Indeed, the OH radicals in the lower troposphere are more efficient to oxidize CH$_4$ molecules, because the CH$_4$+OH reaction rate increases with temperature (Eq. 1). When considering the standard atmosphere, the reaction rate corresponding to the surface temperature of 288K ($5.2 \times 10^{-15}$ s$^{-1}$) is more than twice that for the 500hPa temperature of 253K ($2.2 \times 10^{-15}$ s$^{-1}$). Despite similar volume-weighted tropospheric mean [OH] of ~10.4 $\times 10^5$ molec cm$^{-3}$, MOCAGE simulated much lower CH$_4$ mixing ratios (1275 ppbv) than CMAM (1540 ppbv) and MRI-ESM1r1 (1639 ppbv) because of its higher near surface [OH] ($19 \times 10^5$ molec m$^{-3}$) (Table 4). The spatial distribution of the OH radicals also slightly influences CH$_4$ oxidation. Indeed, the [OH] of EMAC-L90MA are higher than those of CESM-WACCM for both, tropospheric ($11.1 \times 10^5$ versus 10.7 $\times 10^5$ molec cm$^{-3}$) and near-surface ($12.5 \times 10^5$ versus $12.4 \times 10^5$ molec cm$^{-3}$) means, but a slightly higher CH$_4$ burden is found for the former (1579 versus 1575ppbv, Table 6). This is because EMAC-L90MA simulated higher [OH] over the ocean, while CESM-WACCM OH is more concentrated over land closer to CH$_4$ source regions. The model experiments also emphasize that volume-weighted tropospheric concentrations cannot fully indicate the atmospheric oxidizing efficiency for CH$_4$, as has been discussed by Lawrence et al. (2001). Tropospheric mean [OH] weighted by reaction rates with CH$_4$, which consider



both temperature and CH$_4$ distributions, can be a better indicator for CH$_4$ oxidation (Lawrence et al., 2001).


## 4.2 Impacts on CH$_4$ spatial distribution and growth rate

In order to address the question of inter-annual variability of atmospheric CH$_4$, we scaled each OH field globally to get the same CH$_4$ loss (for the year 2000) as the one obtained with INCA NMHC OH field (see Sect. 2.2.2). The scaling used a single global scaling factor (per OH field). As listed in Table 4, after

scaling most OH fields have volume-weighted tropospheric mean concentrations closer to INCA NMHC (9.7 $\times 10^5$ molec cm$^{-3}$), within the range of 9.0-10.4$\times 10^5$ molec cm$^{-3}$. One exception is MOCAGE, with tropospheric mean [OH] scaled to 7.7$\times 10^5$ molec cm$^{-3}$, due to its distinct vertical distribution (Sect.3.2). This scaling of OH makes it possible to start model experiments at the same initial CH$_4$ burden. Although slightly modifying the magnitude of the global mean [OH], this scaling maintained the spatial and

temporal differences and trend over the 2000-2010 period.

### 4.2.1 Spatial distributions of tropospheric CH$_4$ mixing ratios

We used the scaled OH fields to perform simulations between 2000 and 2010. Figure 6 shows the spatial distribution of tropospheric CH$_4$ mixing ratios for the simulation Run_standard (Table. 2, driven by OH

with inter-annual variations) averaged over 2000-2010. Although all simulations started from the same initial conditions and OH fields were scaled to give the same global CH$_4$ loss as INCA NMHC in 2000, using the differently scaled OH fields still generated a spread of 8 ppbv in the 11-year average tropospheric mean CH$_4$ mixing ratios. The LMDz model using the TransCom OH fields (without inter-annual variability) simulated the highest CH$_4$ mixing ratios (1735 ppbv), while the one using the CMAM

OH (with slightly increasing trends during the decade) simulated the lowest (1727 ppbv). Differences between the global tropospheric mean [OH] cannot explain these differences (see Table 4). Clearly the different spatial (horizontal and vertical) and temporal variations of the OH fields (as described in Sect.



3), which were kept in this experiment, significantly modify the simulated CH$_4$ mixing ratios (Table 7 and Fig. 6).


Looking at latitudinal CH$_4$ mixing ratios, the inter-model differences appear larger than in the global mean (Fig. 6 and Table 7). The model spreads of the mean CH$_4$ mixing ratios over 60 °S-90 °S and 60 °N-90 °N range from 1771 to 1794ppbv and 1784 to 1812ppbv, respectively. Here, we define the N/S gradient of CH$_4$ as the difference in mean CH$_4$ mixing ratio between the latitudinal bands 60 °N - 90 °N and from 60 °S

- 90 °S. With the TransCom OH field (N/S ratio =1.0), the model simulated 12-43% larger N/S gradients of CH$_4$ (129 ppbv) than other simulations (90-115 ppbv) driven by OH fields with higher N/S OH ratios of 1.2-1.5. Previous model studies have attributed the overestimation of the CH$_4$ N/S ratio to an underestimation of model inter-hemispheric exchange time (e.g. Zimmermann et al., 2018). Our results show that uncertainties in OH distributions can also contribute to such model biases.


4.2.2 Changes in CH$_4$ mixing ratios

To assess the influence of OH inter-annual variations on CH$_4$ mixing ratios, we calculated the difference in the simulated CH$_4$ between the standard run (Run_standard) and the simulations with fixed [OH] (Run_fix_OH, Table 2). The Run_fix_OH simulations show that global tropospheric mean CH$_4$ mixing

ratios increased by 75 ppbv from 2000 to 2010 (Fig. 7, black dashed lines), due to the enhanced emissions (Fig.1). The increase in [OH] can obviously reduce CH$_4$ growth. An increase in [OH] by 0.1-0.3 $\times10^5$ molec cm$^{-3}$ (1%-3%) (Fig. 7, orange lines) during this period leads to a reduction of the CH$_4$ mixing ratios by 5-15 ppbv by 2010 (Fig. 7, blue lines).The largest reductions are found when using CESM1-WACCM and CMAM OH fields, given the continuous OH growth in these models. Compared to Run_fix_OH, we

estimated that such reductions in CH$_4$ mixing ratios offset 7- 20% of the CH$_4$ increase driven by the rising CH$_4$ emissions of our scenario over the period 2000 to 2010.



After 2010, CCMI REF-C2 experiments simulated increasing, relatively stable, or decreasing OH variations, thus having a variable influence on $CH_4$ variations. Over the period 2011-2016, [OH] simulated by EMEC-L90MA, CESM-WACCM and CMAM stabilizes at a level $0.2$-$0.4 \times 10^5$ molec $cm^{-3}$ higher than the concentrations in 2000, further reducing $CH_4$ mixing ratios by up to 20-30 ppbv in 2016 (Fig. 7, blue lines). Other OH fields have similar concentrations over 2010-2016 as in the early 2000s (Fig. 7, orange lines), thus simulating $CH_4$ mixing ratios that remain close to Run_fix_OH with differences less than a few +/-ppbv.

As large uncertainties remain regarding the inter-annual variations and trend of OH after 2010, we have tested two additional OH scenarios: Run_OH_inc (with an annual increase of 1‰ $yr^{-1}$) and Run_OH_dec (with an annual decrease of 1% $yr^{-1}$), to assess the uncertainty range of the impact of OH changes (the orange areas in Fig. 7) on $CH_4$ mixing ratios (the blue areas in Fig. 7). In these two scenarios, the mean [OH] of run_OH_dec is $\sim 7 \times 10^5$ molec $cm^{-3}$ (7%) lower than run_OH_inc in 2016. If OH decreases at 1% $yr^{-1}$ after 2010, by 2016, the differences in $CH_4$ mixing ratios between Run_OH_dec and Run_fix_OH range -7–30ppbv, with the lower end (-7ppbv) simulated by OH from CESM1-WACCM given its highest [OH] in 2010. On the contrary, Run_OH_inc simulated 3-39 ppbv lower $CH_4$ mixing ratios compared to Run_fix_OH (the blue areas in Fig. 7). As such, uncertainties in the OH trend can clearly lead to $> \pm 30$ ppbv changes in $CH_4$ mixing ratios (the gray areas in Fig. 7) after only 6 years of simulations, as compared to the fix-OH case.

It is now interesting to compare the range of simulated [$CH_4$] changes induced by OH scenarios to changes in surface $CH_4$ observations, in order to quantify how much of the model-observation mismatch could potentially be attributed to uncertainties in [OH] and its variability (Fig. 8). To do so, we used surface $CH_4$ observations from the National Oceanic and Atmospheric Administration (NOAA) networks and selected stations with 17 years continuous records over 2000-2016. Since the simulated absolute $CH_4$



mixing ratios largely depend on the initial conditions and OH fields, we compared changes in the simulated and observed global $CH_4$ mixing ratios starting at the same point in 2000. The observed $CH_4$

shows zero growth between 2000 and 2006 and then increases by 5.6 ppbv $yr^{-1}$ between 2006 and 2012 (6.4 ppb $yr^{-1}$ for 2006-2010) and by 9.4 ppbv $yr^{-1}$ after 2012. In this study we do not expect to fit these $CH_4$ trends as this inter-comparison was not conducted with a set of optimized emissions. It has already been noticed that standard $CH_4$ emission inventories lead to overestimated $CH_4$ mixing ratios (e.g. Saunois et al., 2016). Indeed, neither Run_standard nor Run_fix_OH simulations do capture the

stagnation before 2006, and overestimated surface $CH_4$ increments by 2.5-5.2 ppbv $yr^{-1}$ during the period 2006-2010. Based on Run_fix_OH, on average over 2000-2016 and depending on the OH scenario, we found that [OH] changes can emphasize the model-observation mismatch by up to 19%, or fill the gap between model simulations and observations by up to 54% (figure 8). Such comparisons strongly suggest that a better understanding of OH inter-annual variations and trends is required in order to simulate more

reliable $CH_4$ trends in atmospheric chemistry models. Atmospheric chemistry transport model (Dalsøren et al., 2016) and box model studies (Rigby et al., 2017; Turner et al., 2017) have pointed out that variations in OH can partly explain the recent $CH_4$ trends. However, current top-down estimates of $CH_4$ emissions usually assume constant [OH] (Saunois et al., 2017) and attribute the model-observation discrepancies only to surface emissions rather than changes in [OH]. Our results confirm the potentially significant role

played by the still uncertain OH changes in the actual changes of methane emissions since 2000.

**5 Conclusions**

We have analyzed 14 OH fields (11 from CCMI experiments, 2 from INCA model simulations, 1 from

TransCom) to investigate the inter-model differences in the spatial distributions and trends of tropospheric OH, and estimated the influences of OH spatio-temporal distributions on tropospheric $CH_4$ by feeding them in different simulations with LMDz offline chemistry transport model.



Simulated global volume-weighted tropospheric mean [OH] are within the range of $8.7 \times 10^5$-$12.8 \times 10^5$

molec cm$^{-3}$, which is consistent with the (large) multi-model range of previous estimates. CCMI and

INCA models simulated larger [OH] in the Northern hemisphere than in the Southern hemisphere (N/S

ratio of 1.2-1.5), with consistently high OH values over anthropogenic emission hotspots in North

America, East and South East Asia, while TransCom OH shows a N/S ratio close to 1.0. In the vertical,

TransCom OH reaches its maximum value at about 600 hPa, while CCMI and INCA OH fields either

continuously decrease with altitude or show very small vertical variations in the troposphere. The factors

most likely responsible for these inter-model differences include i) large NO mixing ratios leading to high

surface and mid-tropospheric [OH] (Teyssèdre et al., 2007; Pevell et al. 2018), and ii) different natural

emissions as well as VOC species and chemical mechanisms driving spatial model discrepancies over

natural ecosystems.


Simulated OH fields show small year-to-year variations, within $\pm 0.5 \times 10^5$ molec cm$^{-3}$ during 1960-2010.

From 2000 to 2010, year-to-year variations in OH are smaller than in previous decades and all OH fields

increase by about $0.01$-$0.03 \times 10^5$ molec cm$^{-3}$ yr$^{-1}$. Such an increase in OH is mainly attributed to the

significant positive OH trend over East and Southeast Asia ($>0.1 \times 10^5$ molec cm$^{-3}$ yr$^{-1}$), in response to more

OH production by NO$_x$ than OH loss by CO (Riahi et al., 2011).

The inter-model differences in tropospheric OH burden generate a wide range of CH$_4$ burdens (1204-

1882ppbv) when used to simulate steady-state CH$_4$ mixing ratios in the atmospheric chemistry model

LMDz. Our findings suggest that not only different global mean [OH], but also differences in the

horizontal and vertical distributions between OH fields are responsible for this range (CH$_4$ destruction

rates by OH increase with temperature).

The $CH_4$ simulations for 2000-2016 using OH with inter-annual variation show that inter-model differences of the OH N/S ratio lead to 12-43% differences in the $CH_4$ N/S gradient. For the time period 2000-2010, we found that a 1%-3% increase in [OH] leads to a 5-15 ppb reduction of the $CH_4$ mixing ratio until 2010, accounting for 7-20% of the simulated emission driven $CH_4$ increase over this period. After 2010, the ensemble of OH scenarios tested here leads to differences in $CH_4$ mixing ratio of up to 30 ppb by 2016. Comparing with surface observations, we found that [OH] changes can emphasize the model-observation mismatch by up to 19%, or fill the gap between model simulations and observations by up to 54% (Figure 8). Therefore, addressing the OH variability in $CH_4$ source inversions seems critical to avoid a wrong attribution of $CH_4$ changes to emission changes only. Future work is needed to quantify the impact of this ensemble of OH fields on $CH_4$ emissions obtained by inversion and to generate improved OH fields to be used in $CH_4$ inversion studies.

**Author contributions**

YZ, MS, and PB designed the study, analyzed data and wrote the manuscript. BZ and XL helped with data analysis and model simulations. JC, RJ, and AS discussed the results. ED provided the atmospheric in situ data. Other co-authors provided numerical model outputs. All co-authors commented on the manuscript.

**Acknowledgements**

This work has been funded by the Gordon and Betty Moore Foundation through Grant GBMF5439 'Advancing Understanding of the Global Methane Cycle' and takes benefit of the expertise of the Global Carbon Project methane initiative.

We acknowledge the modelling groups for making their simulations available for this analysis, the joint WCRP SPARC/IGAC Chemistry-Climate Model Initiative (CCMI) for organising and coordinating the model data analysis activity, and the British Atmospheric Data Centre (BADC) for collecting



and archiving the CCMI model output.

The EMAC simulations have been performed at the German Climate Computing Centre (DKRZ) through support from the Bundesministerium für Bildung und Forschung (BMBF). DKRZ and its scientific steering committee are gratefully acknowledged for providing the HPC and data archiving resources for this consortial project ESCiMo (Earth System Chemistry integrated Modelling).

The CESM project is supported primarily by the National Science Foundation.

Eugene Rozanov work is partially supported by Swiss national Science Foundation under grant 200020_182239 (POLE) and the gained information will be used to improve next versions of the CCM SOCOL.

Andrea Stenke was supported by the Swiss National Science Foundation under grant 200021_138037 (FuMES).

UMUKCA-UCAM model integrations were performed using the ARCHER UK National Supercomputing Service and MONSooN system, a collaborative facility supplied under the Joint Weather and Climate Research Programme, which is a strategic partnership between the UK Met Office and the Natural Environment Research Council.

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



# Tables

**Table 1.** List of CCMI models included in this study with model versions and references.[1]

| Model | Version | References |
|---|---|---|
| CESM1-CAM-chem | CCMI_23 | Tilmes et al.(2015, 2016) |
| CESM1-WACCM | CCMI_30 | Solomon et al. (2015); Garcia et al. (2016); Marsh et al. (2013) |
| CMAM | v2.1 | Jonsson et al. (2004); Scinocca et al. (2008) |
| EMAC(offers two resolutions: EMAC-L47MA and EMAC-L90MA) | v2.51 | Jöckel et al. (2010, 2016) |
| GEOSCCM | v3 | Molod et al. (2012, 2015); Oman et al. (2011, 2013); Nielsen et al. (2017) |
| HadGEM3-ES | HadGEM3 GA4.0, NEMO 3.4, CICE, UKCA, MetUM8.2 | Walters et al.(2014); Madec(2008); Hunke and Lipscombe(2008); Morgenstern et al.(2009); O'Connor et al.(2004); Hardiman et al.(2017) |
| MOCAGE | v2.15.1 | Josse et al. (2004); Guth et al. (2016) |
| MRI-ESM1r1 | v1.1 | Yukimoto et al. (2012, 2011); Deushi and Shibata (2011) |
| SOCOL3 | v3 | Revell et al. (2015); Stenke et al. (2013) |
| UMUKCA-UCAM | MetUM 7.3 | Morgenstern et al. (2009); Bednarz et al. (2016) |

[1] The table refers to Table 2 in Morgenstern et al. (2017)

**Table 2.** List of LMDz experiments and model setups.

|  | Simulation period | Inter annual variability in [OH] |
|---|---|---|
| Run_standard | 2000-2010 | 2000-2010 |
| Run_REF-C2 | 2011-2016 | 2010 apply inter-annual variability from REF-C2 |
| Run_OH_inc | 2011-2016 | 2010 apply annual growth rate of 1‰ |
| Run_OH_dec | 2011-2016 | 2010 apply annual decrease rate of 1% |
| Run_fix_OH | 2000-2016 | Constant OH (year 2000) |



**Table 3.** Inter-hemispheric ratios (N/S) of hemispheric mean OH and volume-weighted tropospheric mean [OH] for four latitude bands (in $10^5$ molec.cm$^{-3}$) averaged over the years 2000 to 2010.

| OH fields | N/S ratio | 90°S-30°S ($10^5$ molec.cm$^{-3}$) | 30°S-0° ($10^5$ molec.cm$^{-3}$) | 0°-30°N ($10^5$ molec.cm$^{-3}$) | 30°N-90°N ($10^5$ molec.cm$^{-3}$) |
|---|---|---|---|---|---|
| **TransCom** | 1.0 | 5.8 | 12.7 | 11.8 | 6.2 |
| **INCA NMHC-AER-S** | 1.3 | 4.7 | 10.6 | 12 | 7.5 |
| **INCA NMHC** | 1.2 | 5.7 | 11.9 | 13.4 | 7.8 |
| **CESM1-CAM4Chem** | 1.4 | 5.7 | 12.4 | 15.3 | 9.2 |
| **CESM1-WACCM** | 1.3 | 5.9 | 12.3 | 15.1 | 9.3 |
| **CMAM** | 1.2 | 5.6 | 13.1 | 14.3 | 8.3 |
| **EMAC-L47MA** | 1.2 | 6 | 13.5 | 15.6 | 8.4 |
| **EMAC-L90MA** | 1.2 | 6.3 | 13.8 | 15.7 | 8.6 |
| **GEOSCCM** | 1.2 | 6.5 | 14.8 | 16.8 | 9.1 |
| **HadGEM3-ES** | 1.4 | 4.1 | 10.4 | 12.5 | 8.1 |
| **MOCAGE** | 1.5 | 5.5 | 11.4 | 14.3 | 10.2 |
| **MRI-ESM1r1** | 1.2 | 4.7 | 13.7 | 15.3 | 7.3 |
| **SOCOL3** | 1.5 | 6.8 | 13.5 | 17.0 | 14.0 |
| **UMUKCA-UCAM** | 1.3 | 5.6 | 13.7 | 14.9 | 9.9 |

**Table 4.** Global mean [OH] averaged over the troposphere and three vertical pressure levels (in $10^5$ molec cm$^{-3}$) over the years 2000 to 2010.

| | Tp-v[1] | Tp-m[2] | 750[3] | 500 | 250 | Tp scaled[2] |
|---|---|---|---|---|---|---|
| **TransCom** | 9.1 | 10.0 | 9.9 | 12.8 | 9.2 | 9.5 |
| **INCA NMHC-AER-S** | 8.7 | 9.4 | 11.3 | 10.4 | 7.8 | 9.3 |
| **INCA NMHC** | 9.7 | 10.4 | 11.8 | 11.4 | 8.9 | 9.7 |
| **CESM1-CAM4Chem** | 10.7 | 11.3 | 12.2 | 12.3 | 10.7 | / |
| **CESM1-WACCM** | 10.7 | 11.4 | 12.4 | 12.5 | 10.7 | 9.9 |
| **CMAM** | 10.4 | 11.3 | 14.3 | 11 | 10.5 | 9.3 |
| **EMAC-L47MA** | 10.9 | 11.3 | 12.1 | 12 | 10.3 | / |
| **EMAC-L90MA** | 11.1 | 11.5 | 12.5 | 12.2 | 10.2 | 10.3 |
| **GEOSCCM** | 11.8 | 12.3 | 12.3 | 13.7 | 12 | 10.4 |
| **HadGEM3-ES** | 8.8 | 9.9 | 12.7 | 10.8 | 7.7 | / |
| **MOCAGE** | 10.4 | 12.5 | 19 | 13.5 | 7.7 | 7.7 |
| **MRI-ESM1r1** | 10.3 | 10.6 | 12.2 | 10.4 | 9.4 | 10.2 |



| | | | | | |
|---|---|---|---|---|---|
| **SOCOL3** | 12.8 | 14.4 | 19.4 | 15.1 | 10.9 | 9.0 |
| **UMUKCA-UCAM** | 11.0 | 11.9 | 14.9 | 11.7 | 10.5 | / |

[1] Tp-v refers to the volume-weighted tropospheric mean [OH].

[2] Tp-m refers to the mass-weighted tropospheric mean [OH]

[3] 750 refers to the volume-weighted average from the surface to 750hPa, 500 refers to the volume-weighted average from 750hPa to 500 hPa, and 250 refers to the volume-weighted average from 500 to 250hPa.

[4] Tp scaled refer to the volume-weighted global tropospheric mean [OH] after scaling to the same $CH_4$ loss as with INCA NMHC in 2000.

**Table 5.** Global volume-weighted mean $O^{1D}$ photolysis rate, specific humidity, CO and NO mixing ratios averaged over the whole troposphere and three pressure altitude levels for CCMI models over 2000 to 2010.[1]

| | $O^{1D}$ photolysis rate $10^5$ s$^{-1}$ | | | | Specific humidity ($10^{-3}$ g/kg) | | | |
|---|---|---|---|---|---|---|---|---|
| | 750 | 500 | 250 | Tp[2] | 750 | 500 | 250 | Tp |
| **CESM1-CAM4Chem** | 0.9 | 1.3 | 1.5 | 1.3 | 11.7 | 4.2 | 0.9 | 4.7 |
| **CESM1-WACCM** | 1.0 | 1.3 | 1.6 | 1.3 | 11.6 | 4.1 | 0.9 | 4.7 |
| **CMAM** | 0.9 | 1.2 | 1.3 | 1.1 | 10.1 | 3.5 | 0.8 | 4 |
| **EMAC-L47MA** | 0.7 | 1 | 1.3 | 1.1 | 11.8 | 4.1 | 0.8 | 4.7 |
| **EMAC-L90MA** | 0.7 | 1 | 1.3 | 1.1 | 11.6 | 4.2 | 0.8 | 4.7 |
| **GEOSCCM** | 0.6 | 0.8 | 0.9 | 0.8 | 11.4 | 4.5 | 1.1 | 4.9 |
| **MOCAGE** | / | / | / | / | 11.0 | 3.7 | 0.8 | 4.3 |
| **MRI-ESM1r1** | 0.7 | 0.9 | 1.1 | 0.9 | 11.8 | 4.2 | 0.9 | 4.8 |
| **SOCOL3** | 0.7 | 0.8 | 0.9 | 0.8 | 11.6 | 4.6 | 1.0 | 4.8 |
| | **CO ppbv** | | | | **NO pptv** | | | |
| | 750 | 500 | 250 | Tp | 750 | 500 | 250 | Tp |
| **CESM1-CAM4Chem** | 76 | 71 | 70 | 71 | 9 | 4 | 12 | 13 |
| **CESM1-WACCM** | 75 | 70 | 69 | 70 | 9 | 5 | 12 | 12 |
| **CMAM** | 77 | 68 | 64 | 69 | 17 | 4 | 17 | 26 |
| **EMAC-L47MA** | 85 | 77 | 70 | 75 | 8 | 4 | 11 | 14 |
| **EMAC-L90MA** | 84 | 76 | 69 | 74 | 8 | 5 | 11 | 17 |
| **GEOSCCM** | 78 | 74 | 73 | 74 | 9 | 5 | 13 | 13 |
| **MOCAGE** | 67 | 68 | 67 | 67 | 26 | 14 | 17 | 20 |
| **MRI-ESM1r1** | 93 | 86 | 83 | 86 | 10 | 5 | 20 | 32 |
| **SOCOL3** | 79 | 73 | 74 | 74 | 48 | 10 | 14 | 25 |

[1] HadGEM3-ES and UMUKCA-UCAM are not analyzed since model output has been regridded to too





coarse vertical pressure levels.

[2] Tp refers to the total troposheric average, 750 refers to the average from the surface to 750hPa, 500 refers to the average from 750hPa to 500hPa, and 250 refers to the average from 500hPa to 250hPa.

**Table 6.** Global mean tropospheric $CH_4$ mixing ratios as simulated by LMDz using different OH fields and repeating year 2000 over 30 times.

| $CH_4$ mixing ratio (ppbv) | | | |
|---|---|---|---|
| INCA NMHC-AER-S | 1822 | CESM1-WACCM | 1575 |
| **TransCom** | 1776 | **CMAM** | 1540 |
| **INCA NMHC** | 1709 | **GEOSCCM** | 1503 |
| **MRI-ESM1r1** | 1693 | **MOCAGE** | 1275 |
| **EMAC-L90MA** | 1579 | **SOCOL3** | 1204 |

**Table 7.** LMDz simulated $CH_4$ mixing ratios (in ppbv) averaged over each latitudinal band and the years
2000 to 2010 simulated from the standard experiment (Run_standard) using different OH fields.

| | 90 °S-60 °S | 60 °S-0 ° | 0 °-60 °N | 60 °N-90 °N | N/S gradient[1] |
|---|---|---|---|---|---|
| **TransCom** | 1683 | 1697 | 1769 | 1812 | 129 |
| INCA NMHC-AER-S | 1687 | 1698 | 1757 | 1795 | 108 |
| **INCA NMHC** | 1687 | 1700 | 1762 | 1802 | 115 |
| CESM1-WACCM | 1688 | 1701 | 1757 | 1794 | 106 |
| **CMAM** | 1682 | 1694 | 1756 | 1796 | 114 |
| EMAC-L90MA | 1685 | 1698 | 1759 | 1798 | 113 |
| **GEOSCCM** | 1688 | 1701 | 1764 | 1803 | 115 |
| **MOCAGE** | 1686 | 1699 | 1753 | 1788 | 102 |
| **MRI-ESM1r1** | 1691 | 1702 | 1762 | 1803 | 112 |
| **SOCOL3** | 1694 | 1707 | 1754 | 1784 | 90 |

[1] N/S gradient is defined as the difference between 60 °N to 90 °N and 60 °S to 90 °S.



# Figures

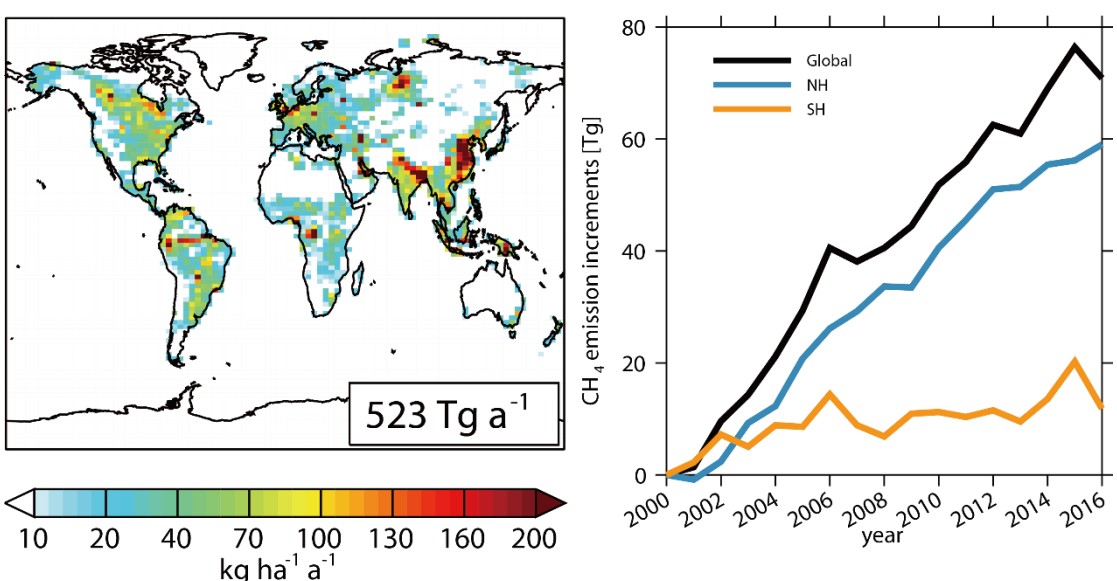

**Figure 1.** Spatial distribution of global $CH_4$ emissions averaged between 2000 and 2016 (left) and a time series of $CH_4$ emissions relative to year 2000 emissions (482 Tg $CH_4$ $a^{-1}$) (right) for the globe (black line), Northern hemisphere (NH, blue line) and Southern hemisphere (SH, yellow line), respectively.

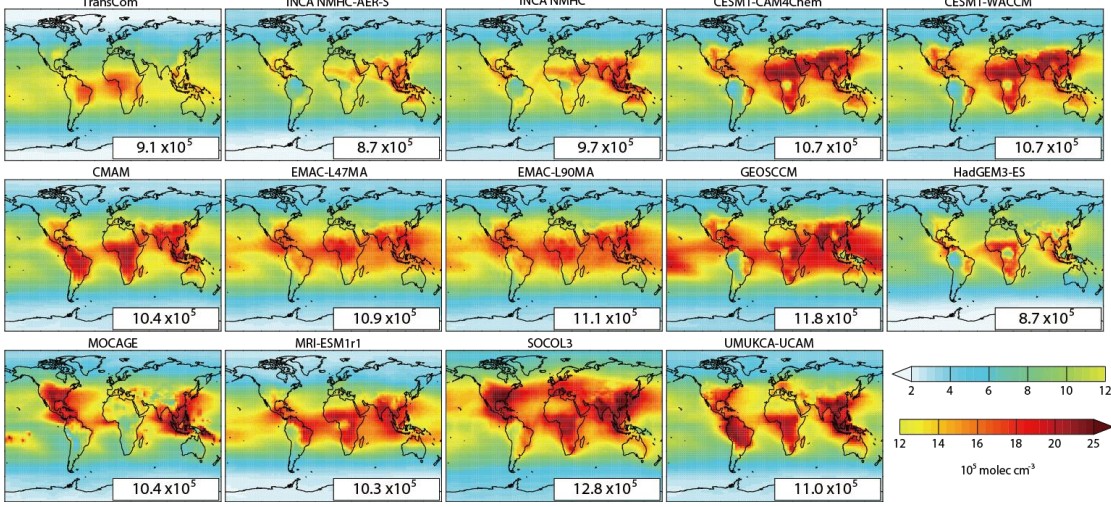

**Figure 2.** The spatial distributions of volume-weighted tropospheric mean OH fields of TransCom, INCA, and CCMI models averaged for 2000-2010. Global mean values ($10^5$ molec $cm^{-3}$) are shown as insets.




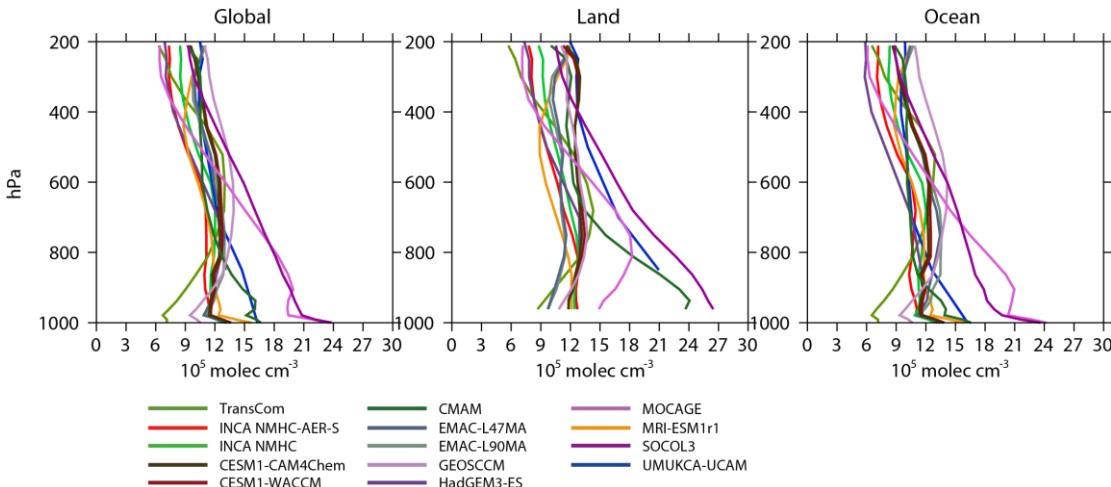

**Figure 3.** Vertical distributions of [OH] averaged over the globe (left), land (middle) and ocean (right) for 2000-2010.

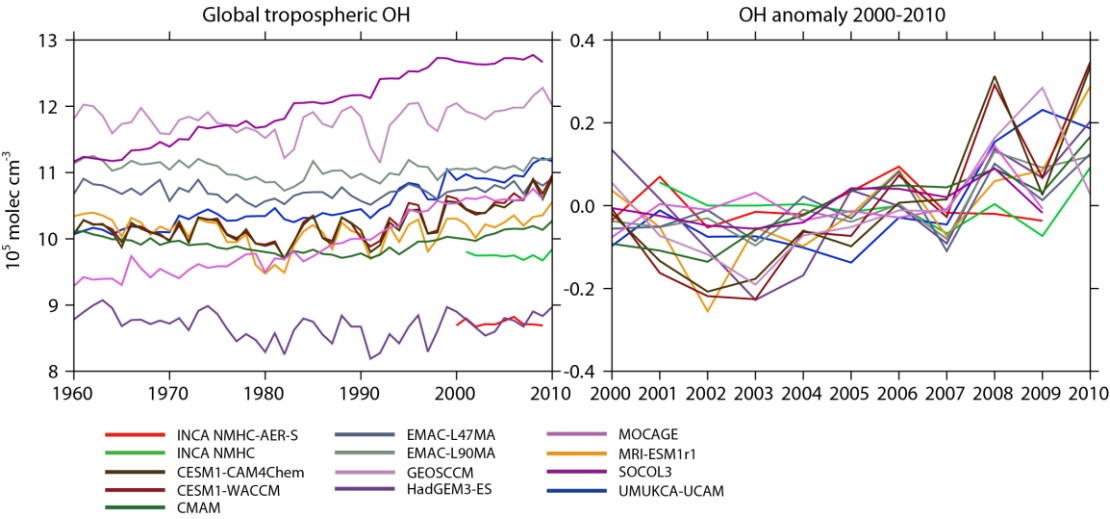

**Figure 4.** Left: Inter-annual variations of global volume-weighted tropospheric mean [OH] from CCMI and INCA model simulations from 1960 to 2010. Right: OH anomaly during 2000-2010, in reference to the mean concentration over the period 2000-2010 for each model.



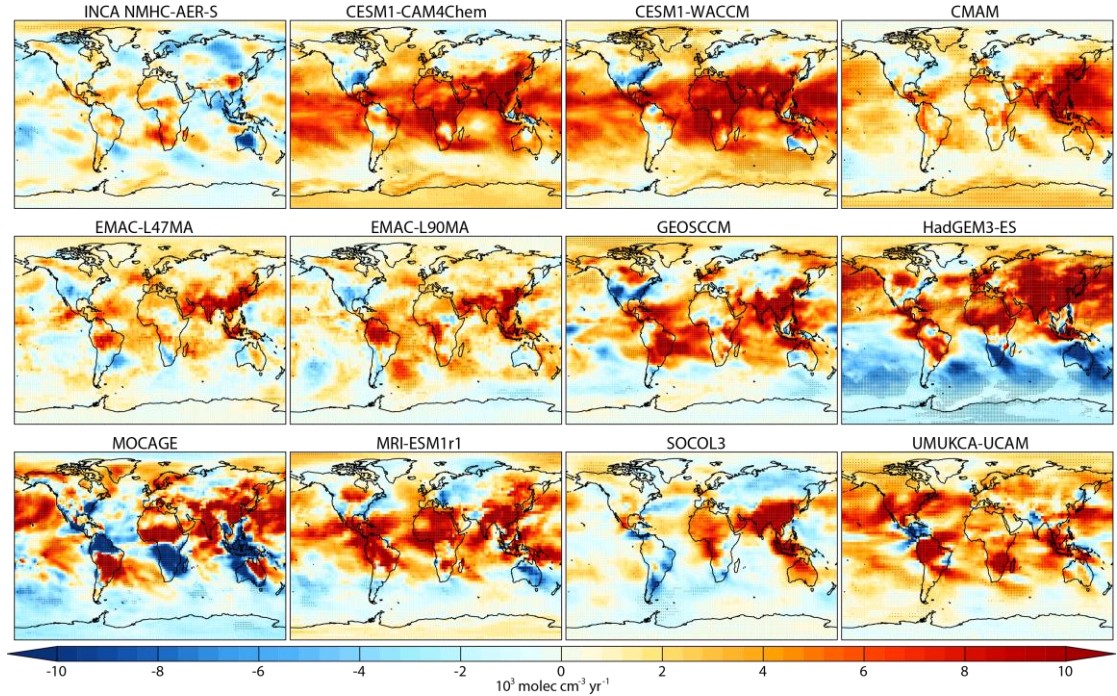

**Figure 5.** Spatial distribution of tropospheric OH trends from 2000 to 2010 (in $10^3$ molec cm$^{-3}$ yr$^{-1}$). Black
dots denote model grid-cells with statistically significant trends (p-value $< 0.05$)

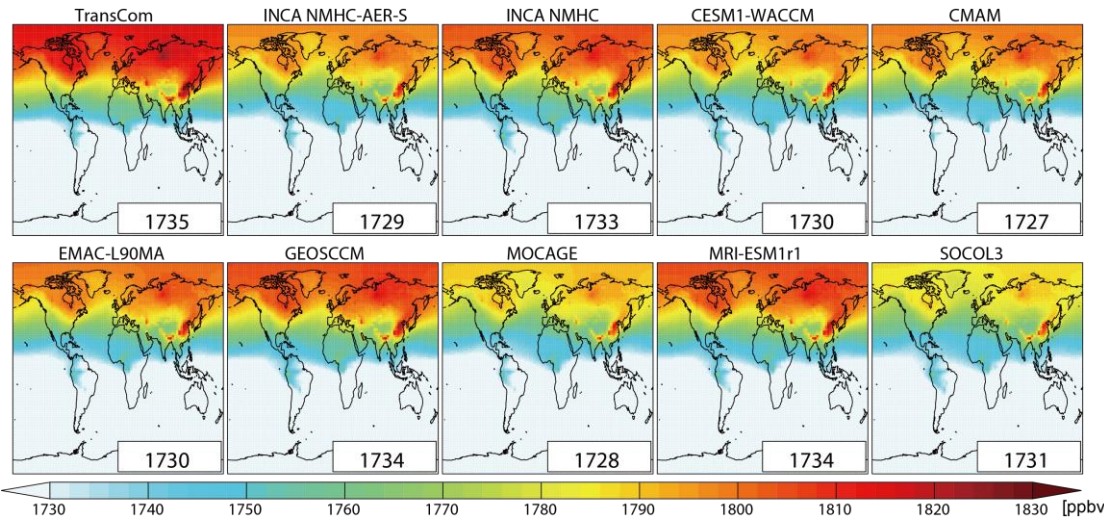

**Figure 6.** Spatial distribution of volume-weighted tropospheric mean CH$_4$ mixing ratios averaged from
2000 to 2010 as simulated by LMDz with different OH fields in the LMDz model. The global mean values
in units of ppbv are shown as insets.



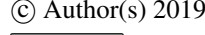

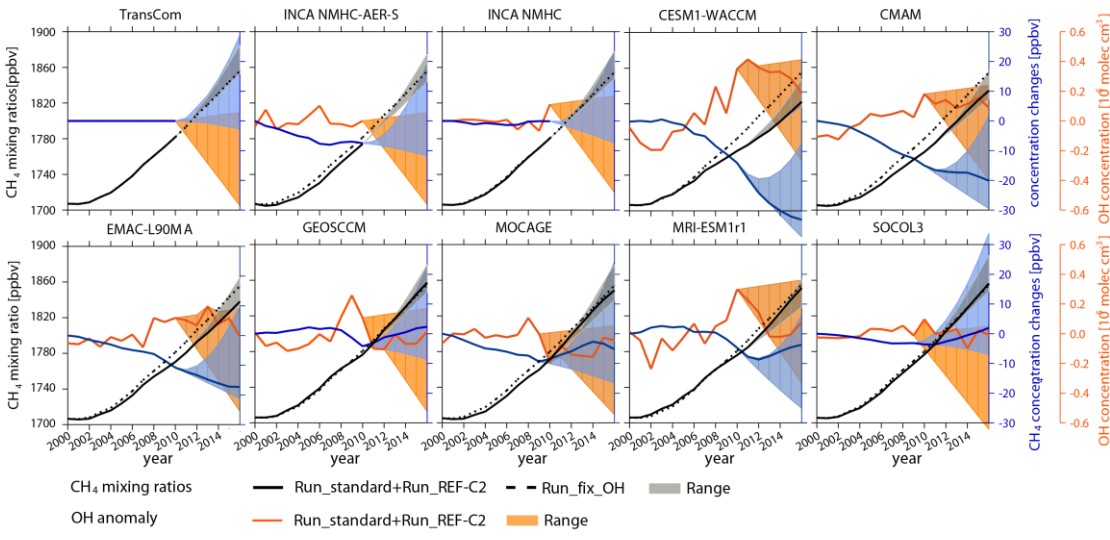

**Figure 7.** Time-series of global tropospheric $CH_4$ mixing ratios and [OH] associated with the model experiments listed in Table 2. The black lines represent the evolution of $CH_4$ mixing ratios with varying (solid lines) or with constant (dashed lines) OH. The varying OH case is obtained using OH inputs from Run_standard from 2000 to 2010 followed by Run_REF-C2 from 2011 to 2016 (see Table 2). The blue solid lines represent the corresponding differences between the simulations with varying OH and with constant OH. The orange solid line represents the corresponding anomalies in tropospheric [OH] (with the average over 2000-2010 as reference). The shaded areas correspond to the range obtained from all simulations over 2010-2016 (Table 2) for tropospheric $CH_4$ mixing ratios (grey), for changes in tropospheric $CH_4$ mixing ratios (blue) and for changes in tropospheric [OH] (orange).





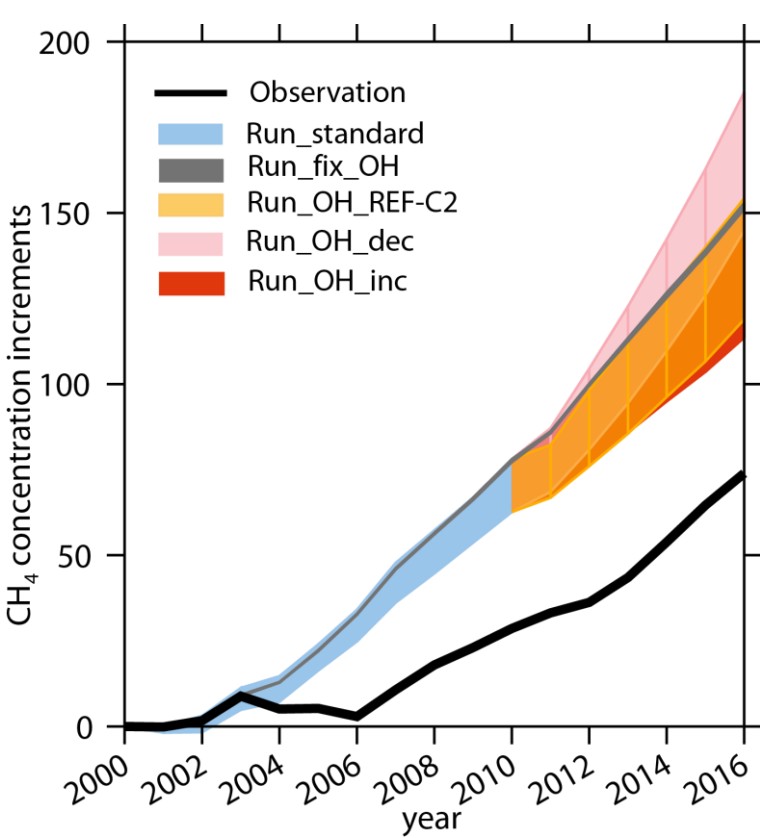

**Figure 8.** Time series of surface CH$_4$ mixing ratio increments compared to 2000 for NOAA observations (black line) and model ranges from all the LMDz experiments collected at observation sites (shades) and described in the text and in Table 2.
