# Peer review of "Inter-model comparison of global hydroxyl radical (OH) distributions and their impact on atmospheric methane over the 2000-2016 period"

_Atmospheric Chemistry and Physics, 2019_

## Short Comment (SC1) · 10 Apr 2019

For certain experiments in this study, the authors use a compilation of methane emissions that is based on bottom-up estimates (inventories and process-models) and not constrained by atmospheric observations (lines 263-266). The resulting increase in emissions between 2000 and 2016 is 70 Tg/yr. This is a large difference, compared to emission scenarios constrained by methane mole fractions [CH4], which place the increase in the order of 20-40 Tg/yr (depending on start and end period). For examples, see Saunois et al. (2017, doi.org/10.5194/acp-17-11135-2017) with best estimates of around 24 Tg/yr between the two periods 2002-2006 and 2008-2012 or Nisbet et

al. (2019, doi.org/10.1029/2018GB006009), who estimate a ∼44 Tg/yr difference between 2000-2005 and 2015-2018. Bottom up emissions have been repeatedly shown to overestimate the increase in methane after 2007, as reviewed by Saunois et al. (2017). Consequently, the increase of [CH4] in the atmosphere is strongly overestimated in the present study as seen in Fig. 8, where the modelled difference in [CH4] between 2000 and 2010 is >70 ppb, while the observation is ∼25 ppb.

The modelling presented here is outside my area of expertise, yet it would be interesting how the overestimate in methane emissions will influence the simulated CH4-OH dynamics. E.g., is the offline LMDz model subject to CH4-feedback on OH? Would a lower rate of emissions increase produce a significantly different result?

The unrealistic CH4 evolution makes it difficult to assess the importance of the findings for the recent methane budget. For example, the authors state that varying OH from 2000 to 2010 suppressed [CH4] by 5-15 ppb (line 538). Would that value hold for a slower [CH4] increase? Does the stated OH effect as equivalent to 7-20% of the emissions change (line 540) represent a fixed percentage of any emissions increase or would it scale with the emissions scenario (in which case the OH effect could be equivalent to 16-45% of the emissions change of Saunois et al., 2017)?

In my opinion, the relevance of the presented findings for the wider community could be strongly enhanced by a more realistic emission scenario.

---

## Referee Comment (RC1) · Vaishali Naik (Referee) · 2 May 2019

Review of "Inter-model comparison of global hydroxyl radical (OH) distributions and their impact on atmospheric methane over the 2000-2016 period" by Zhao et al for ACP

This study presents a multi-model comparison of the distribution and trends in hydroxyl (OH) radical and an analysis of how the different OH fields influence atmospheric methane trends over the 2000-2016 period. This paper is very relevant to the current debate in the scientific literature about the drivers of recent methane increase in general and specifically the role of OH in this renewed methane growth since 2007. Generally, the paper is well-organized and addresses scientific questions within the scope of ACP.

The authors have analyzed the free-running coupled chemistry-climate model simulations derived from the CCMI project as well as from three additional models not participating in CCMI. Like the ACCMIP models, the models considered here also produce a wide range of OH distributions and agree on the sign of change in OH over the 1980 to the present (2010) time period. The authors largely attribute the reasons for intermodel diversity in the simulated mean OH distributions to differences in the representation of nitrogen oxide, and natural emissions of NMVOCs and their chemistry. I felt that the analysis could be a bit deeper, especially the role of differences in lightning NOx and stratospheric ozone could be investigated. The authors also do not evaluate the models against proxy observational constraints (such as ozone, water vapor, CO, NOx column, methane lifetime) to assess their skill in simulating OH levels. How do we know which model is closer to reality? I acknowledge that this is a difficult question to answer but feel that some effort is needed to evaluate the models. Finally, the authors attribute the increasing trend in tropical OH over 2000 to 2010 to increasing NOx emissions, again without an in-depth quantitative analysis. I have highlighted these and other issues in my specific comments below. I recommend the publication of this paper after it has been revised to include additional evaluation and analysis.

Specific Comments:

L70: Levy (1972) is the wrong reference here. The correct reference should be Levy (1971).

L71: Atomic excited oxygen is denoted as O($^1$D). Please revise O$^{1D}$ throughout the manuscript.

L99-103: The tropospheric methane chemical lifetime against OH loss from ACCMIP models was calculated as the global annual mean **atmospheric** methane burden divided by annual mean methane tropospheric loss by OH. Please revise.

L112-114: "The precise scenario…" sounds odd. Please revise to "The precise reasons for stagnation and renewed growth of methane still remain unclear…"

L127-130: With their chemistry model, Dalsoren et al (2016) actually simulate an ~8% increase in OH over the 1970 to 2012 time period. Not only is this trend as large as that inferred by Turner et al and Rigby et al, but is also in a completely opposite direction and is also contrary to

the no OH trend deduced by Nicely et al (2018). I think the point that there is a mismatch in the OH trends inferred from observations and simulated by global models should be explicitly highlighted.

L141-144: The configuration of ACCMIP simulations was not ideal for assessing interannual variability in OH due primarily to the fact that these simulations were not performed continuously from year to year but were timeslices with emissions as well meteorology characteristic of the decade being simulated. So in that sense, the ACCMIP simulations did allow one to assess decadal (or multi decadal) but not year to year variability. I think this sentence should be revised to something like this for clarity: "Year-to-year integrations of CCMI and INCA models driven by time-varying emissions and meteorology facilitate the investigation of interannual variability in OH which was not possible using the ACCMIP time-slice simulations"

L158-159: From the perspective of understanding the influence of OH changes on methane, I think the analysis of REF-C1SD is more relevant because it will likely reproduce (or be close to) observed changes in various climate related factors that influence OH (e.g., humidity, temperature) as well as chemical composition since it is nudged to observed meteorology. I would highly encourage the analysis of REF-C1SD simulations if results from at least 5 of the models analyzed here are available.

L168-169: Please clarify what is tropospheric ozone chemistry coupled to - stratospheric chemistry, physical climate?

L172-173: Define the chemical names before using them HCHO (formaldehdye) and C5H8 (isoprene).

L177-178: I think it would be helpful to clarify that after year 2000, the MACCity emissions (therefore REF-C1 simulation) follow the RCP8.5 scenario. It would also be helpful to clarify whether biomass burning emissions are also from MACCity and if they vary from year to year.

L190: According to the description in Morgenstern et al (2017), methane concentrations prescribed in the CCMI models vary in time following the RCP6.0 contrary to the specification (global mean of ~1750ppbv averaged from over 2000-2010) described here. Please clarify.

Section 2.1: It would be helpful if the authors could clarify how tropopause is defined (in order to calculate tropospheric OH from the models) somewhere in this section.

Section 2.2.1: The motivation for using this model and simulations became clear to me only after I read section 2.2.2. I think the motivation for using this model with methane emissions should be clarified up front in this section.

L232: access should be replaced with assess.

L238-239: It is mentioned above that the offline LMDz5B is run with OH field from CCMI models. Does it also use O(1D) fields from the CCMI models?

L245-248: Please clarify which chemistry module is used in the setup here.

L301, L552: Please confirm if this is indeed 1‰ yr-1 or 1% yr-1.

L313-314: Revise "...which overestimation of [OH]..."

L317: IPCC (2011) is missing from the reference list. Also suggest citing the IPCC chapter (though I do not think there is a 2011 IPCC report relevant here) rather than the whole report.

Tables 3, 4, 5, and 7: It would be helpful to provide a quantitative measure of model spread (e.g., coefficient of variation or range) at the end of the columns which would make it easier for the reader to quickly get an estimate of intermodel diversity.

L336-347: In addition to individual model maps, it would be helpful to have a map of the standard deviation of OH concentrations across the models considered here to clearly see the regions of high/low spread. Also, could variability in lightning NOx emissions across models be a cause of the model spread in OH? What about differences in simulated stratospheric ozone across the models; relevant for southern hemisphere OH differences? I think a deeper analysis of the factors that influence OH is needed to assess the reasons for the spread in OH fields.

L356: Is it GEOSGCM or GEOSCCM?

L390-410: Murray et al. (2014) show that lightning NOx plays a key role in controlling OH as also mentioned on lines 414 to 417. How different/similar are the models in their representation of the vertical distribution of lightning NOx emissions?

Figure 3 would also benefit from plot of standard deviation across models for each region.

L430-431: What is the mean year to year variation over this period in units of percent?

L432-434: Are the numbers in parentheses the changes in OH concentrations from 1960 to 1980? If so, it would be helpful to provide percent changes as well.

L447-450: How do the year-to-year variations in OH from CCMI and INCA models compare with the results of Turner et al. (2017), Rigby et al. (2017) and Nicely et al. (2018) using different approaches?

L455-458: During this time period, there have been increases in water vapor as well (e.g., Dessler and Davis 2010) that could potentially influence OH trends in the tropical regions. Admittedly, the increase in NOx emissions has been significant, but I am not sure if the analysis presented here can be used to say, with confidence, that the strong positive trend in OH is

being solely driven by NOx emissions. It would be helpful to perform some regression analysis to build confidence in the conclusions here.

L459-460: Changes in stratospheric ozone also dominate in the southern hemisphere and could potentially add to inter-model differences.

L478-485: The sensitivity of CH4 oxidation due to OH to lower tropical tropospheric temperature has been established by prior studies (see John et al., 2012 and references therein), some of which should be cited here. Additionally, the discussion here will benefit from a table of CH4 loss flux simulated by LMDz using the different OH fields for tropics (30S-30N), northern mid to high lat (30N-90N), and southern mid to high (30S-90S) and for three vertical levels. Or these numbers could be plotted in the form of Lawrence plots as in Lawrence et al. (2001) for each OH field. This would clearly show the diversity in the spatial distribution of methane loss resulting from the different OH fields.

L497-499: Is the scaling applied to every year over the 2000-2010 period or just for year 2000?

L510-513: I am not sure if it is surprising that there was a spread in the simulated methane distributions across models, particularly, because the scaling was performed on a global scale (matching the global methane loss flux) rather than at the grid-cell level. While the global OH may match with INCA NMHC OH field after scaling but the spatial distribution may still be different producing differences in the simulated global mean methane distributions. .

L534-535: Is this true for all the models shown in figure 7? From the figure, methane for Run_fix_Oh and run_standard seem to overlap for INCA NMHC and GEOSCCM models.

L545: It should be EMAC-L90MA.

L544-547: Add "relative to the Run_fix_OH" after "....further reducing CH4 mixing ratios by up to 20-30 ppbv in 2016…"

L565: Have the measurements been combined in a specific way to create global mean? Are the model CH4 values sampled at the location of these stations?

L569-571: Reference Figure 8 here.

L574-576: From Figure 8, it looks like that the three lines (black obs, blue Run_standard and grey Run_fix_OH) are overlapping until about 2003. y -axis of Figure 8 also needs concentration units (ppb?)

L576-578: I am not sure if I understand this sentence (especially "fill the gap between model simulations and observations by up to 50%"). Could the authors please clarify and also how these percentages have been calculated.

L609-610: This assertion needs to be substantiated or toned down in the absence of more detailed analysis (due to lack of diagnostics such as OH prod and loss). I believe the Riahi et al 2011 reference is not appropriate here as it documents RCP8.5 emissions but does not comment on chemistry-composition impacts from changes in these emissions.

References:

Dessler, A. E., and S. M. Davis (2010), Trends in tropospheric humidity from reanalysis systems,J. Geophys. Res.,115, D19127, doi:10.1029/2010JD014192.

John, J. G., Fiore, A. M., Naik, V., Horowitz, L. W., and Dunne, J. P.: Climate versus emission drivers of methane lifetime against loss by tropospheric OH from 1860–2100, Atmos. Chem. Phys., 12, 12021-12036, https://doi.org/10.5194/acp-12-12021-2012, 2012.

Levy, H.II (1971), Normal atmosphere: Large radical and formaldehyde concentrations predicted, Science, 173, 141–143.

Nicely, J. M., Canty, T. P., Manyin, M., Oman, L. D., Salawitch, R. J., Steenrod, S. D., et al. (2018). Changes in global tropospheric OH expected as a result of climate change over the last several decades. Journal of Geophysical Research: Atmospheres, 123, 10,774– 10,795. https://doi.org/10.1029/2018JD028388

Rigby M, et al. Role of atmospheric oxidation in recent methane growth. Proc Natl Acad Sci USA. 2017;114:5373–5377

Turner AJ, Frankenberg C, Wennberg PO, Jacob DJ. Ambiguity in the causes for decadal trends in atmospheric methane and hydroxyl. Proc Natl Acad Sci USA. 2017;114:5367–5372.

---

## Referee Comment (RC2) · Anonymous Referee #2 · 24 Jun 2019

General comments: The manuscript "Inter-model comparison of global hydroxyl radical (OH) distribution and their impact on atmospheric methane over the 2000-2016 period" written by Yuanhong Zhao describes the inter-model differences in spatial distribution and temporal evolution of OH concentrations, and elucidates the impacts of simulated OH concentration fields on CH4 using the LMDz chemical transport model. The manuscript contains novel investigation to reveal inter-annual variations in OH and its impact on CH4 over recent decades using multi-model approach. The topic of the manuscript is certainly within the scope of ACP. Overall, the manuscript is well written and easy to follow. I would like to consider the publication of the manuscript from ACP, while I have several comments below which should be addressed before publication.

[Figure]

Specific comments: 2.1. OH field Is the prescribed biogenic NMVOC emissions (p. 8, l. 187) climatology? Please clarify.

How did the authors prescribe the ECLIPSE and RCP85 emission inventories in the INCA simulations during the periods before 2004, between 2006-2009, and after 2011?

2.2.2. Model simulations Please clarify how the OH increasing and decreasing rates are determined in the $Run_OH_inc and Run_OH_dec simulations. Why are the rates +1 and -1$

3.1. Spatial distributions of tropospheric OH The authors attributed possible causes of too large interhemispheric differences in OH in the CCMI models to model O3 and CO biases and unaccounted processes in some of the CCMI models, as reported by previous studies. Why is not the model performance on O3 and CO in the CCMI ensembles evaluated or referred? It might be better to cite Strode et al. (2016), Revell et al. (2018), and other papers.

3.3. Factors contributing to inter-model differences Why the authors did not assess inter-model differences in tropospheric O3 burden? The tropospheric O3 burden should also affect primary production of OH.

Do inter-model differences in vertical distribution of lighting NO production affect OH vertical distributions?

3.4. Inter-annual variations of OH What is possible cause of significant positive [OH] trends over the tropics (p. 19, l. 454)?

4.2.1. Spatial distributions of tropospheric CH4 mixing ratio Could you explain how inter-model differences in spatial and temporal OH variations affect the simulated global CH4 mixing ratio more in depth?

Technical corrections: p. 13, l. 317: publication year is missing.

p. 19, l. 461: "and" is typo?

[Figure]

[Figure]

p. 23, l. 545: typo for EMAC-L90MA?

Reference Revell, L. E., Stenke, A., Tummon, F., Feinberg, A., Rozanov, E., Peter, T., Abraham, N. L., Akiyoshi, H., Archibald, A. T., Butchart, N., Deushi, M., Jöckel, P., Kinnison, D., Michou, M., Morgenstern, O., O'Connor, F. M., Oman, L. D., Pitari, G., Plummer, D. A., Schofield, R., Stone, K., Tilmes, S., Visioni, D., Yamashita, Y., and Zeng, G.: Tropospheric ozone in CCMI models and Gaussian process emulation to understand biases in the SOCOLv3 chemistry–climate model, Atmos. Chem. Phys., 18, 16155-16172, https://doi.org/10.5194/acp-18-16155-2018, 2018.

Strode, S. A., Worden, H. M., Damon, M., Douglass, A. R., Duncan, B. N., Emmons, L. K., Lamarque, J.-F., Manyin, M., Oman, L. D., Rodriguez, J. M., Strahan, S. E., and Tilmes, S.: Interpreting space-based trends in carbon monoxide with multiple models, Atmos. Chem. Phys., 16, 7285-7294, https://doi.org/10.5194/acp-16-7285-2016, 2016.

---

## Author Response (AR1)

*Reply to SC1: 'Short comment on Zhao et al. (2019)'*

**We thank the reviewer for the helpful comments provided. All of them have been addressed in the revised manuscript. Please see our itemized responses below.**

**Comment 1:** For certain experiments in this study, the authors use a compilation of methane emissions that is based on bottom-up estimates (inventories and process-models) and not constrained by atmospheric observations (lines 263-266). The resulting increase in emissions between 2000 and 2016 is 70 Tg/yr. This is a large difference, compared to emission scenarios constrained by methane mole fractions [CH4], which place the increase in the order of 20-40 Tg/yr (depending on start and end period). For examples, see Saunois et al. (2017, doi.org/10.5194/acp-17-11135-2017) with best estimates of around 24 Tg/yr between the two periods 2002-2006 and 2008-2012 or Nisbet et al. (2019, doi.org/10.1029/2018GB006009), who estimate a ~44 Tg/yr difference between 2000-2005 and 2015-2018. Bottom up emissions have been repeatedly shown to overestimate the increase in methane after 2007, as reviewed by Saunois et al. (2017). Consequently, the increase of [CH4] in the atmosphere is strongly overestimated in the present study as seen in Fig. 8, where the modelled difference in [CH4] between 2000 and 2010 is >70 ppb, while the observation is ~25 ppb. The modelling presented here is outside my area of expertise, yet it would be interesting how the overestimate in methane emissions will influence the simulated CH4-OH dynamics.

**Comment 3:** Would a lower rate of emissions increase produce a significantly different result? The unrealistic CH4 evolution makes it difficult to assess the importance of the findings for the recent methane budget. For example, the authors state that varying OH from 2000 to 2010 suppressed [CH4] by 5-15 ppb (line 538). Would that value hold for a slower [CH4] increase? Does the stated OH effect as equivalent to 7-20% of the emissions change (line 540) represent a fixed percentage of any emissions increase or would it scale with the emissions scenario (in which case the OH effect could be equivalent to 16-45% of the emissions change of Saunois et al., 2017)? In my opinion,

the relevance of the presented findings for the wider community could be strongly enhanced by a more realistic emission scenario.

**Response for these two related comments: We performed two additional experiments with emissions fixed to 2000 to test the influence of emission scenarios on the results presented.**

**We have added in the text:**

**Section 2.2.2 (method)**

 **"In addition, we conducted two simulations during 2000-2010 driven by emission inventories fixed to the year 2000 to test the influences of the emission bias on our results. The two simulations use OH fields simulated by CESM-WACCM, one with inter-annual variations of OH (Run_fix_emis) and the other one with OH field fixed to 2000 (Run_fix_emis_OH)."**

**Section 4.2.2 (results)**

**"To test whether the impacts of [OH] year-to-year variations on $CH_4$ mixing ratios depends on the chosen emission scenarios, we compare the above results with that calculated by an extreme scenario where model simulations are driven by fixed emissions (year 2000, Run_fix_emis and Run_fix_emis_OH, table 2). With emissions fixed to 2000, the $CH_4$ mixing ratio increased by 2ppbv from 2000 to 2010, and increasing OH (CESM-WACCM OH fields) can reduce $CH_4$ mixing ratio by 13.5ppb in 2010, comparable to 13.9 ppb calculated by Run_std and Run_fix_OH with CESM_WACCM OH fields. The results indicate only small effect of emission scenario choice on the absolute changes in $CH_4$ mixing ratios due to OH variations. However, our choices have a large effect on the relative changes to the total modeled $CH_4$ increase. Indeed, if we use the emission scenarios that match observations (~+25ppbv of $CH_4$ mixing ratio increase from 2000-2010 instead of ~70 ppb here, Ed Dlugokencky, NOAA/ESRL, 2019), the $CH_4$ mixing ratio changes due to OH can contribute to more than half (13.5-13.9ppbv versus 25ppbv) of the changes driven by emissions."**

**We also included two model experiments in Table 2.**

**Table 2.** List of LMDz experiments and model setups.

| | Simulation period | Inter annual variability in [OH] | Inter annual variability in CH4 emissions |
|---|---|---|---|
| Run_standard | 2000-2010 | 2000-2010 | 2000-2010 |
| Run_REF-C2 | 2011-2016 | 2010 apply inter-annual variability from REF-C2 | 2011-2016 |
| Run_OH_inc | 2011-2016 | 2010 apply annual growth rate of 1‰ | 2011-2016 |
| Run_OH_dec | 2011-2016 | 2010 apply annual decrease rate of 1% | 2011-2016 |
| Run_fix_OH | 2000-2016 | Constant OH (year 2000) | 2010-2016 |
| Run_fix_emis | 2000-2010 | 2000-2010(CESM-WACCM only) | Constant (2000) |
| Run_fix_emis_oh | 2000-2010 | Constant OH (year 2000 CESM-WACCM only) | Constant (2000) |

**Comment 2:** E.g., is the offline LMDz model subject to CH4-feedback on OH?

**Response: We clarify in the text:" Chemical sinks of CH$_4$ are calculated using prescribed three-dimensional OH and O($^1$D) fields, and variation in CH$_4$ cannot feedback on OH."**

*Reply to RC1: 'Review of Zhao et al. (2019)'*

**We thank the reviewer for the helpful comments. All of them are addressed and answered below.**

**Comment:** This study presents a multi-model comparison of the distribution and trends in hydroxyl (OH) radical and an analysis of how the different OH fields influence atmospheric methane trends over the 2000-2016 period. This paper is very relevant to the current debate in the scientific literature about the drivers of recent methane increase in general and specifically the role of OH in this renewed methane growth since 2007. Generally, the paper is well-organized and addresses scientific questions within the scope of ACP.

The authors have analyzed the free-running coupled chemistry-climate model simulations derived from the CCMI project as well as from three additional models not participating in CCMI. Like the ACCMIP models, the models considered here also produce a wide range of OH distributions and agree on the sign of change in OH over the 1980 to the present (2010) time period. The authors largely attribute the reasons for intermodel diversity in the simulated mean OH distributions to differences in the representation of nitrogen oxide, and natural emissions of NMVOCs and their chemistry. I felt that the analysis could be a bit deeper, especially the role of differences in lightning NOx and stratospheric ozone could be investigated. The authors also do not evaluate the models against proxy observational constraints (such as ozone, water vapor, CO, NOx column, methane lifetime) to assess their skill in simulating OH levels. How do we know which model is closer to reality? I acknowledge that this is a difficult question to answer but feel that some effort is needed to evaluate the models. Finally, the authors attribute the increasing trend in tropical OH over 2000 to 2010 to increasing NOx emissions, again without an in-depth quantitative analysis. I have highlighted these and other issues in my specific comments below. I recommend the publication of this paper after it has been revised to include additional evaluation and analysis.

**Response:**
**In this paper our aim is to estimate the impact of OH distribution in space and time on methane changes since 2000 using an ensemble of state-of-the-art atmospheric models. We acknowledge that the depth of analysis of the root causes of what we find here can be increased the lack of evaluation of these models in our paper. This is because finding the "best model" is very difficult, regarding**

the multiple criteria to take into account even when only looking at OH, that we use such an ensemble. Doing a full evaluation of these models is beyond the scope of our study but to better explain what we find and stretghten a bit model evaluation, we now compare ozone simulated by the CCMI models with TOMS/SBUV observations (Fig. S4) and we have calculated tropospheric chemical lifetime in table 4. In addition, the CO column and tropospheric $O_3$ column have already been evaluated by Strode et al. (2016) and Revell et al. (2019), we also cite these two references in the text.

We have added in the text:

" The tropospheric chemical $CH_4$ lifetime of the models that provided $CH_4$ chemical loss data are 8.7±1.1 yr. Both the multi-model mean and the (large) range of [OH] as well as tropospheric $CH_4$ chemical lifetime are consistent with previous multi-model results given by the ACCMIP project (Naik et al., 2013; Voulgarakis et al., 2013), as well as with inversions based on MCF observations (Bousquet et al., 2005; Rigby et al., 2017)."

"Previous studies have attributed the inconsistency between the simulated and the observed OH N/S ratios to a model overestimation of $O_3$ and underestimation of CO over the Northern Hemisphere (Naik et al., 2013; Young et al., 2013; Strode et al., 2015), which also have been reported for CCMI models (Strode et al., 2016; Revell et al., 2018), …"

  All of other more specific comments have been addressed in the revised manuscript. Please see out itemized responses below.

**Specific Comments:**

**Comments:** L70: Levy (1972) is the wrong reference here. The correct reference should be Levy (1971).
**Response: Thanks for pointing it out. The reference is now corrected.**

**Comments:   L71:** Atomic excited oxygen is denoted as O(1D). Please revise $O^{1D}$ throughout the manuscript.
**Response: Changed as suggested**

L99-103: The tropospheric methane chemical lifetime against OH loss from ACCMIP models was calculated as the global annual mean **atmospheric** methane burden divided by annual mean methane

tropospheric loss by OH. Please revise.

**Response: Changed as suggested**

**Comments:** L112-114: "The precise scenario…" sounds odd. Please revise to "The precise reasons for stagnation and renewed growth of methane still remain unclear…"

**Response: Changed as suggested**

**Comments:** L127-130: With their chemistry model, Dalsoren et al (2016) actually simulate an ~8% increase in OH over the 1970 to 2012 time period. Not only is this trend as large as that inferred by Turner et al and Rigby et al, but is also in a completely opposite direction and is also contrary to the no OH trend deduced by Nicely et al (2018). I think the point that there is a mismatch in the OH trends inferred from observations and simulated by global models should be explicitly highlighted.

**Response: We add in the text "Meanwhile, not only the OH trend calculated by atmospheric chemistry models cannot reach consensus, but it can also be different from the OH trend inferred by top-down approaches from observations. Indeed, Dalsøren et al. (2016) simulated ~ 8% increase in OH during 1970 to 2012, while other models mostly calculated only a small increase of [OH] (decrease in CH4 lifetime) or no trend in [OH] from 1980s to 2000s (e.g. Voulgarakis et al., 2013; Nicely et al., 2018). Top-down observation-constrained approaches (e.g. Rigby et al., 2017) tend to find flat to decreasing OH trend over this period but with larger year-to-year variations than models."**

**Comments:** L141-144: The configuration of ACCMIP simulations was not ideal for assessing interannual variability in OH due primarily to the fact that these simulations were not performed continuously from year to year but were timeslices with emissions as well meteorology characteristic of the decade being simulated. So in that sense, the ACCMIP simulations did allow one to assess decadal (or multi decadal) but not year to year variability. I think this sentence should be revised to something like this for clarity: "Year-to-year integrations of CCMI and INCA models driven by time-varying emissions and meteorology facilitate the investigation of interannual variability in OH which was not possible using the ACCMIP time-slice simulations"

**Response: "Changed as suggested"**

**Comments:** L158-159: From the perspective of understanding the influence of OH changes on methane, I think the analysis of REF-C1SD is more relevant because it will likely reproduce (or be close to) observed changes in various climate related factors that influence OH (e.g., humidity, temperature) as well as chemical composition since it is nudged to observed meteorology. I would highly encourage the analysis of REF-C1SD simulations if results from at least 5 of the models analyzed here are available.

**Response:**

**We add in the main text:" The models of REF-C1SD experiment are nudged towards reanalysis datasets. The REF-C1SD experiment is not analyzed in the main text since it has been conducted by only part of the models and covers a shorter time period. A comparison of spatial and vertical distributions of OH fields from REF-C1 experiment with that from REF-C1SD reveals only small latitudinal differences (<10%, see Section S1)."**

**We have added Section S1 with table S1, S2, and Fig. S1 in the supplemental:**

**"S1 OH fields from CCMI REF-C1 experiments.**

**We compare spatial and vertical distributions of OH fields from REF-C1 (main text) with that from REF-C1SD to access influences from dynamic biases. Of CCMI models included in this study, 7 models conducted REF-C1SD experiments (EMAC offers fields at two different model resolutions). Fig. S1 shows the spatial distributions of the volume-weighted tropospheric mean [OH] averaged from 2000 to 2010 simulated by REF-C1SD experiments, Table S1 summarizes their inter-hemispheric ratios and mean values over four latitudinal bands. The volume-weighted mean [OH] averaged over the troposphere and over three pressure latitudinal intervals are calculated in Table S2. By comparing Fig. S1, table S1, and table S2 with Fig. 2, table 3, and table 4, respectively, we find that OH fields from REF-C1 and REF-C1SD experiments show similar spatial and vertical distributions. Only CESM and MOCAGE simulated recognizable different N/S ratios (small differences within 0.1-0.2) by REF-C1 and REF-C1SD experiments, and the differences in mean OH over four latitudinal bands and latitudinal intervals are within 10%."**

**Table S1.** Inter-hemispheric ratios (N/S) of hemispheric mean OH and volume-weighted tropospheric

mean [OH] for four latitude bands (in $10^5$ molec.cm$^{-3}$) averaged over the years 2000 to 2010 from CCMI REF-C1SD experiment.

| OH fields | N/S ratio | 90°S-30°S ($10^5$ molec.cm$^{-3}$) | 30°S-0° ($10^5$ molec.cm$^{-3}$) | 0°-30°N ($10^5$ molec.cm$^{-3}$) | 30°N-90°N ($10^5$ molec.cm$^{-3}$) |
|---|---|---|---|---|---|
| CESM1-CAM4Chem | 1.3 | 6.3 | 13.3 | 15.9 | 8.7 |
| CESM1-WACCM | 1.2 | 6.6 | 13.3 | 15.9 | 9 |
| CMAM | 1.2 | 5.8 | 12.8 | 13.7 | 8.1 |
| EMAC-L47MA | 1.2 | 6.4 | 14.1 | 15.6 | 8.5 |
| EMAC-L90MA | 1.2 | 6.2 | 13.5 | 15.1 | 8.4 |
| MOCAGE | 1.3 | 6.1 | 12.1 | 14.5 | 8.9 |
| MRI-ESM1r1 | 1.2 | 4.7 | 14.2 | 15.7 | 6.9 |
| UMUKCA-UCAM | 1.3 | 5.6 | 13.9 | 15.2 | 10.1 |

**Table S2.** Global mean [OH] averaged over the troposphere and three vertical pressure levels (in $10^5$ molec cm$^{-3}$) over the years 2000 to 2010 from CCMI REF-C1SD experiment.

| | Tp[1] | 750 | 500 | 250 |
|---|---|---|---|---|
| CESM1-CAM4Chem | 11.1 | 12.1 | 13.1 | 11.5 |
| CESM1-WACCM | 11.2 | 12.3 | 13.4 | 11.8 |
| CMAM | 10.1 | 14.3 | 10.9 | 10.8 |
| EMAC-L47MA | 11.2 | 12.4 | 12.4 | 11.1 |
| EMAC-L90MA | 10.9 | 12.3 | 12.1 | 10.2 |
| MOCAGE | 10.4 | 19.2 | 15 | 7.3 |
| MRI-ESM1r1 | 10.5 | 12.4 | 10.8 | 9.7 |
| UMUKCA-UCAM | 11.2 | 16.0 | 12.4 | 10.6 |

[1] Tp refers to the volume-weighted tropospheric mean [OH], 750 refers to the volume-weighted average from the surface to 750hPa, 500 refers to the volume-weighted average from 750hPa to 500 hPa, and 250 refers to the volume-weighted average from 500 to 250hPa.

[Figure]

**Figure S1.** The spatial distributions of volume-weighted tropospheric mean OH fields CCMI REF-C1SD experiments averaged for 2000-2010 . Global mean values ($10^5$ molec cm$^{-3}$) are shown as insets.

**Comments:** L168-169: Please clarify what is tropospheric ozone chemistry coupled to - stratospheric chemistry, physical climate?

**Response: We change "coupled" to "detailed"**

**Comments:** L172-173: Define the chemical names before using them HCHO (formaldehyde) and C5H8 (isoprene).

**Response: Thanks for pointing it out. We add the chemical names as suggested.**

**Comments:** L177-178: I think it would be helpful to clarify that after year 2000, the MACCity emissions (therefore REF-C1 simulation) follow the RCP8.5 scenario. It would also be helpful to clarify whether biomass burning emissions are also from MACCity and if they vary from year to year.

**Response: We have clarified this by adding :**
**L185: "….(which follow the RCP8.5 inventory after 2000),…" after the REF-C1 experiment continued to use the MACCity inventory.**
**L188: "Biomass burning emissions used in REF-C1 are RETRO inventory (Schultz et al. 2008) before 1996 and GFEDv3 inventory (van der Werf et al., 2010) for 1997-2010 with interannual variability."**

[Figure]

**Figure S2. Multi-model mean (left), standard deviation(middle). and standard deviation relative to multi-model mean of tropospheric mean OH fields shown in figure 2.**

**We cite this figure by: "We further assessed the simulated OH spread by comparing the detailed spatial distributions of OH fields in Fig. 2 and Fig.S2"**

**We also add in the text L375:**

**" Tropospheric mean [OH] over the Amazon forest show large variations of >5.0 ×10$^5$ molec cm$^{-3}$, representing more than 50% to the multi-model mean(Fig.S2). In a more diffuse way, high latitudes of the northern hemisphere also contribute to model spread (25-35% of the model mean, Fig. S2). Besides these, inter-model differences also exist over the open ocean (up to 25% of the model mean, Fig.S2)."**

(2) To answer about the "could variability in lightning NO$_x$ emissions across models be a cause of the model spread in OH", we have added table S3 in the supplement:

Table S3. Lighting NOx emission (Tg N yr$^{-1}$) over three pressure altitudinal intervals and the total troposphere of CCMI models over 2000-2010.

|  | Surface-750hPa | 750-500hPa | 500-250hPa | 250-100hPa | tp |
|---|---|---|---|---|---|
| CMAM | 0.7 | 0.4 | 1.5 | 1.7 | 4.2 |
| EMAC-L90MA | 0.2 | 0.5 | 1.3 | 1.8 | 3.7 |
| CESM1-WACCM | 0.2 | 0.6 | 2.7 | 0.7 | 4.2 |
| GEOSCCM | 0.2 | 1.3 | 3.3 | 0.8 | 5.6 |
| MOCAGE | 0.3 | 1.2 | 2.4 | 1.0 | 4.8 |
| MRI-ESM1r1 | 1.4 | 0.7 | 3.2 | 5.2 | 10.2 |
| SOCOL3 | 0.2 | 0.8 | 2.1 | 1.4 | 4.4 |

We also add in the main text:

L431-L435: "Lighting NO$_x$, which are mainly emitted in the middle and upper troposphere, can contribute to inter-model differences in NO and OH distributions (Murray et al., 2013; 2014). We compare lighting NO$_x$ emissions calculated by CCMI models in Table S3. High lighting NOx emissions simulated by MRI-ESM1r1 above 250hPa can explain high NO mixing ratios and increasing OH with altitude over the upper troposphere for this model (Fig. 3). However, High NO in the lower troposphere simulated by MOCAGE and SOCOL3 are not corresponding to high lighting NO$_x$ emissions in these models."

L451 "Lighting NOx emissions range from 3.7-10.2 Tg yr$^{-1}$(table S3)"

The above text about lighting NO$_x$ emissions also response to comments on L390-410.

(3)"What about differences in simulated stratospheric ozone across the models; relevant for southern hemisphere OH differences?"

To analyse influences on southern hemisphere OH, we have added table S5, which compares stratosphere ozone and O($^1$D) photolysis rate for four latitude bands, and also figure S4, which compares total ozone column with satellite observations in the supplement.

**Table S5.** Tropospheric mean stratosphere ozone and O($^1$D) photolysis rate for four latitudinal bands averaged over 2000 to 2010. Multi-model means and standard deviations (Mean ± stand. dev.) are also shown.

| | Stratosphere ozone | | | | O($^1$D) photolysis rates ($10^{-5}$ s$^{-1}$) | | | |
|---|---|---|---|---|---|---|---|---|
| | 90°S-30°S | 30°S-0° | 0°-30°N | 30°N-90°N | 90°S-30°S | 30°S-0° | 0°-30°N | 30°N-90°N |
| CESM1-CAM4Chem | 272 | 222 | 225 | 300 | 0.8 | 1.8 | 1.8 | 0.7 |
| CESM1-WACCM | 261 | 219 | 223 | 286 | 0.8 | 1.9 | 1.8 | 0.7 |
| CMAM | 269 | 228 | 230 | 293 | 0.8 | 1.6 | 1.6 | 0.6 |
| EMAC-L47MA | 298 | 232 | 232 | 299 | 0.6 | 1.5 | 1.5 | 0.6 |
| EMAC-L90MA | 291 | 233 | 233 | 293 | 0.7 | 1.5 | 1.5 | 0.6 |
| GEOSCCM | 249 | 216 | 219 | 286 | 0.6 | 1.2 | 1.1 | 0.4 |
| HadGEM3-ES | 282 | 245 | 248 | 297 | / | / | / | / |
| MOCAGE | 212 | 224 | 245 | 280 | / | / | / | / |
| MRI-ESM1r1 | 280 | 238 | 238 | 301 | 0.6 | 1.3 | 1.3 | 0.5 |
| SOCOL3 | 277 | 238 | 238 | 297 | 0.6 | 1.1 | 1.1 | 0.5 |
| UMUKCA-UCAM | 241 | 236 | 236 | 256 | / | / | / | / |
| Mean ± stand. dev. | 267±25 | 230±9 | 233±9 | 289±13 | 0.7±0.1 | 1.5±0.3 | 1.5±0.3 | 0.6±0.1 |

[Figure]

**Figure S4.** Monthly total column ozone bias from CCMI simulations averaged over 2000-2010 compared to satellite measurements from Total Ozone Mapping Spectrometer/solar backscatter ultraviolet (TOMS/SBUV) (model minus measurement).

**We also add in the main text:**

**"The stratospheric ozone can contribute to inter-model OH discrepancies through influencing $O(^1D)$ photolysis rates. However, we find that models that simulated lower stratosphere and total ozone column are not corresponding to higher $O(^1D)$ photolysis rates and [OH] (table S5 and Fig. S4), since differences in the photolysis schemes coupled to CCMI models can also influence the calculation of $O(^1D)$ photolysis rates (Sukhodolov et al., 2016)."**

**Reference: Sukhodolov, T., Rozanov, E., Ball, W.T., Bais, A., Tourpali, K., Shapiro, A.I., Telford, P., Smyshlyaev, S., Fomin, B., Sander, R., Bossay, S., Bekki, S., Marchand, M., Chipperfield, M.P., Dhomse, S., Haigh, J.D., Peter, T., Schmutz, W., 2016. Evaluation of simulated photolysis rates and their response to solar irradiance variability. Journal of Geophysical Research: Atmospheres 121, 6066-6084.**

**Comments:** L356: Is it GEOSGCM or GEOSCCM?

**Response: We change "GEOSGCM" to "GEOSCCM", thanks for point out the typo.**

**Comments:** L390-410: Murray et al. (2014) show that lightning NOx plays a key role in controlling OH as also mentioned on lines 414 to 417. How different/similar are the models in their representation of the vertical distribution of lightning NOx emissions?

**Response: We answer this comment in the response to comments on L336-347.**

**Comments:** Figure 3 would also benefit from plot of standard deviation across models for each region.

**Response: We calculated the standard deviation in both figure 3 and figure S3 as suggested.**

[Figure]

**Figure 3.** Vertical distributions of [OH] averaged over the globe (left), land (middle) and ocean (right) for 2000-2010. Color lines represent [OH] from individual model simulations, black lines represent multi-model mean values and grey shades represent the standard deviations.

[Figure]

**Figure S3.** Vertical distribution of [OH] averaged over four latitude bands and over the years 2000 to 2010. Color lines represent [OH] from individual model simulations, black lines represent multi-model mean values and grey shades represent the standard deviations.

**Comments:** L430-431: What is the mean year to year variation over this period in units of percent?

**Response: We add then value as suggested:**

**"During this period, all OH fields show small year-to-year variations of 1.9±1.2%, remaining within ±0.5×10⁵ molec cm⁻³."**

**Comments:** L432-434: Are the numbers in parentheses the changes in OH concentrations from 1960 to 1980? If so, it would be helpful to provide percent changes as well.

**Response: We add percent changes as suggested.**

**"For example, [OH] continuously decrease in the CMAM and HadGEM3-ES simulations (~-0.3×10⁵ molec cm⁻³; -3.4%); and increase in SOCOL3 (~+0.6×10⁵ molec cm⁻³; +4.5%), UMUKCA-UCAM (~+0.5×10⁵ molec cm⁻³; +4.8%), and MOCAGE (~+0.5×10⁵ molec cm⁻³; +4.8%) during this 1960-1980,…"**

**Comments:** L447-450: How do the year-to-year variations in OH from CCMI and INCA models compare with the results of Turner et al. (2017), Rigby et al. (2017) and Nicely et al. (2018) using different approaches?

**Response: We including the comparisons by re-organizing this paragraph (see also answer to comment** L127-130)

**"Previous atmospheric chemistry model studies have concluded that anthropogenic activities lead to only a small perturbation of the OH burden, as the increased OH production tend to be compensated by an increased loss through reactions with CO and CH₄ (Lelieveld et al., 2000; Naik et al., 2013). By combining factors that influencing OH, Nicely et al. (2018) modeled a small inter-annual variability of 1.6% during 1980-2015. The year-to-year variations of most CCMI and INCA OH fields are consistent with Nicely et al. (2018), but much smaller than the OH inter-annual variability based on MCF observations (e.g. Bousquet et al., 2005; Montzka et al., 2011), which can reach 8.5±1.0% from 1980 to 2000 (Bousquet et al., 2005), and 2.3±1.5% from 1998 to**

2007(Montzka et al., 2011), as compared to 2.1±0.8% and 1.0±0.5% here for these two periods. As for OH trend, the ensemble of ACCMIP models simulated large divergent OH changes (even in their signs) from 1850 to 2000, but revealed a consistent and significant increase of 3.5±2.2% from 1980 to 2000 (Naik et al., 2013). Here, for the same period the increase of CCMI [OH] is 4.6±2.4%, consistent with the ACCMIP project (Naik et al., 2013) and with other atmospheric chemistry model studies (Dentener et al., 2003; John et al., 2012; Holmes et al., 2013; Dalsøren et al., 2016). The slightly increasing [OH] after 2000 inferred here as well as previous model simulations (e.g. Nicely et al., 2018) cannot help to explain stalled and renewed $CH_4$ growth during the 2000s, as opposed to the decreasing [OH] from mid-2000s calculated by Rigby et al. (2017) and Turner et al (2107) based on MCF observations. "

**Comments:** L455-458: During this time period, there have been increases in water vapor as well (e.g., Dessler and Davis 2010) that could potentially influence OH trends in the tropical regions. Admittedly, the increase in $NO_x$ emissions has been significant, but I am not sure if the analysis presented here can be used to say, with confidence, that the strong positive trend in OH is being solely driven by $NO_x$ emissions. It would be helpful to perform some regression analysis to build confidence in the conclusions here.

**Response: We have calculated the trend of stratospheric O3, specific humidity, CO and NOx emissions in each grid cell for the CCMI models to access the contribution of each factor to OH trends. We cannot do the regression analysis here since only part of the model provide these data and we focus on the spatial distribution of the trends.**

**We have added figure S6a, figure S6b, and figure S6c in the supplement, and in the main text:**
**" By comparing spatial distribution of OH trend with specific humidity (Fig.S6a), $NO_x$ and CO emissions (Fig. S6b), and stratospheric $O_3$ (Fig.S6c), we find that positive OH trend over tropical regions are mainly corresponding to increasing water vapor (Fig. S6a) while faster $NO_x$ emission increases (>5% $yr^{-1}$) than CO (<2% $yr^{-1}$) are consistent with positive OH trend over East and Southeast Asia (Fig. S6b).**
**And**
**"CMAM and HadGEM3-ES show significant increasing and decreasing OH trend over the Antarctic region, respectively, consistent with the significant changes found for stratospheric $O_3$ in these models (Fig. S6c)."**

[Figure]

**Figure S6a.** Spatial distribution of tropospheric specific humidity trends from 2000 to 2010 (in $10^{-2}$ g/kg year$^{-1}$). Black dots denote model grid-cells with statistically significant trends (p-value < 0.05).

[Figure]

**Figure S6b.** Spatial distribution of NO$_x$ (top panels) and CO (bottom panels) trend from 2000 to 2010 (in %). Black dots denote model grid-cells with statistically significant trends (p-value < 0.05).

[Figure]

**Figure S6c.** Spatial distribution of stratosphere $O_3$ column trends from 2000 to 2010 (in DU year$^{-1}$). Black dots denote model grid-cells with statistically significant trends (p-value < 0.05).

**Comments:** L459-460: Changes in stratospheric ozone also dominate in the southern hemisphere and could potentially add to inter-model differences.

**Response: see previous answer : "CMAM and HadGEM3-ES show significant increasing and decreasing OH trend over the Antarctic region, respectively, consistent with the significant changes found for stratospheric $O_3$ in these models (Fig. S6c)."**

**Comments:** L478-485: The sensitivity of CH4 oxidation due to OH to lower tropical tropospheric temperature has been established by prior studies (see John et al., 2012 and references therein), some of which should be cited here. Additionally, the discussion here will benefit from a table of CH4 loss flux simulated by LMDz using the different OH fields for tropics (30S-30N), northern mid to high lat (30N-90N), and southern mid to high (30S-90S) and for three vertical levels. Or these numbers could be plotted in the form of Lawrence plots as in Lawrence et al. (2001) for each OH field. This would clearly show the

diversity in the spatial distribution of methane loss resulting from the different OH fields.

**Response:We have added table S6 in the supplement:**

**Table S6.** CH$_4$ loss by OH oxidation (unit: Tg yr$^{-1}$) as simulated by LMDz using different OH fields and repeating year 2000 over 30 times.

| Run name | | TransCom | INVSAT | INCA | CESM1-WACCM | CMAM | EMAC-L90MA | GEOSCCM | MOCAGE | MRI-ESM1r1 | SOCOL3 |
|---|---|---|---|---|---|---|---|---|---|---|---|
| Surface-750hPa | 30-90 N | 42.9 | 58 | 53.1 | 56 | 56 | 51 | 50.2 | 70.2 | 54.2 | 79.7 |
| | 0-30 N | 90.5 | 106.5 | 105.1 | 101.9 | 115 | 106.2 | 93.8 | 123.7 | 111.6 | 112.9 |
| | 0-30 S | 77.9 | 85.1 | 83.5 | 74.6 | 89.9 | 79.6 | 75.2 | 91.7 | 85.4 | 77.4 |
| | 30-90 S | 16.7 | 16.3 | 18.6 | 18.5 | 18.2 | 18.1 | 17.7 | 24.3 | 16.5 | 20.7 |
| 750-500hPa | 30-90 N | 25.8 | 25.9 | 25.7 | 28 | 22.9 | 26.8 | 26.4 | 26.1 | 22.6 | 31.1 |
| | 0-30 N | 66.8 | 56.4 | 57.5 | 59.5 | 49.5 | 57.9 | 63.5 | 51.1 | 54.4 | 49.1 |
| | 0-30 S | 61 | 45.6 | 46.3 | 45 | 38.9 | 44.4 | 49.9 | 34.6 | 42.8 | 35.5 |
| | 30-90 S | 15.1 | 10.6 | 12.2 | 11.2 | 9.6 | 13 | 11.7 | 9.2 | 9 | 10.5 |
| 500-250hPa | 30-90 N | 9.9 | 11.5 | 11.9 | 13.8 | 11.9 | 12.3 | 12.6 | 8.8 | 11.2 | 12.8 |
| | 0-30 N | 26.1 | 21.9 | 23.3 | 27.4 | 26.4 | 25.7 | 31 | 16.7 | 26.9 | 19.7 |
| | 0-30 S | 24.7 | 17.7 | 19.2 | 20.7 | 21.5 | 20.2 | 23.6 | 11.5 | 20.8 | 14.4 |
| | 30-90 S | 7 | 5.6 | 6.2 | 5.8 | 5.4 | 6.1 | 6 | 3 | 4.8 | 4.5 |

**We also add in the text:" Previous studies have demonstrated that the sensitivity of CH$_4$ oxidation to lower tropical temperature (Spivakovsky et al., 2000; John et al., 2012), and our simulations show that 36%-46% of CH4 is oxidized over lower tropical region (surface-750hPa, 30 S-30 N) (Table S6 )."**

Comments: L497-499: Is the scaling applied to every year over the 2000-2010 period or just for year 2000?

**Response: we clarify by adding in the text:" The single global scaling factor (per OH field) for the year 2000 is applied to every year between 2000 and 2010."**

Comments: L510-513: I am not sure if it is surprising that there was a spread in the simulated methane distributions across models, particularly, because the scaling was performed on a global scale (matching the global methane loss flux) rather than at the grid-cell level. While the global OH may match with INCA NMHC OH field after scaling but the spatial distribution may still be different producing differences in the simulated global mean methane distributions.

**Response: The global scaling approach is what methane inverse modelers usually do. We have applied the same approach here, indeed aiming at assessing how the difference in OH spatial distribution can influence CH$_4$ spatial distributions.**

L545: It should be EMAC-L90MA.

**Response: Thank you for pointing out the typo, we change as suggested.**

Comments: L544-547: Add "relative to the Run_fix_OH" after "....further reducing CH4 mixing ratios by up to 20-30 ppbv in 2016…"

**Response: Changed as suggested.**

Comments: L565: Have the measurements been combined in a specific way to create global mean? Are the model CH4 values sampled at the location of these stations?

**Response: We clarify by adding in the text "The modeled surface CH$_4$ mixing ratios are sampled according to station locations."**

Comments: L569-571: Reference Figure 8 here.

**Response: Changed as suggested.**

Comments: L574-576: From Figure 8, it looks like that the three lines (black obs, blue Run_standard and grey Run_fix_OH) are overlapping until about 2003. y -axis of Figure 8 also needs concentration units (ppb?)

**Response: We change "Indeed, neither Run_standard nor Run_fix_OH simulations do capture the stagnation before 2006" to "Indeed, neither Run_standard nor Run_fix_OH simulations do capture the stagnation during 2004- 2006". And we add the unit to figure 8 as suggested.**

[Figure]

**Figure 8.** Time series of surface $CH_4$ mixing ratio increments compared to 2000 for NOAA observations (black line) and model ranges from all the LMDz experiments collected at observation sites (shades) and described in the text and in Table 2.

**Comments:** L576-578: I am not sure if I understand this sentence (especially "fill the gap between model simulations and observations by up to 50 %"). Could the authors please clarify and also how these percentages have been calculated.

**Response: We clarify by adding in the text "We define highest $CH_4$ mixing ratios simulated by different OH as $CH_{4-H}$, lowest CH4 mixing ratios as $CH_{4-L}$, and CH4 simulated by Run_fix_OH as $CH_{4-fix\_OH}$. Based on Run_fix_OH, on average over 2000-2016 and depending on the OH scenario, we found that [OH] changes can emphasize the model-observation mismatch by up to 19% (mean values of ($CH_{4-H}$－$CH_{4-fix\_OH}$ )/( $CH_{4-fix\_OH}$- observed $CH_4$) during 2000-2016), or limit the model-observation mismatch by up to 54% (mean values of ($CH_{4-fix\_OH}$－$CH_{4-fix\_L}$ )/( $CH_{4-fix\_OH}$- observed $CH_4$.) during 2000-2016) (figure 8)."**

**Comments:** L609-610: This assertion needs to be substantiated or toned down in the absence of more detailed analysis (due to lack of diagnostics such as OH prod and loss). I believe the Riahi et al 2011 reference is not appropriate here as it documents RCP8.5 emissions but does not comment on chemistry-composition impacts from changes in these emissions.

**Response: We have changed the text here based on the new analysis of emissions, water vapor, and ozone column trend now provided in the paper. "Such an increase in OH is mainly attributed to the significant positive OH trend over East and Southeast Asia ($>0.1 \times 10^5$ molec cm$^{-3}$ yr$^{-1}$) in response to more OH production by NO$_x$ than OH destruction by CO, and over tropical regions in response to increasing water vapor. "**

*Reply to RC2: 'Review comments on acp-2019-281'*

**General comments:** The manuscript "Inter-model comparison of global hydroxyl radical (OH) distribution and their impact on atmospheric methane over the 2000-2016 period" written by Yuanhong Zhao describes the inter-model differences in spatial distribution and temporal evolution of OH concentrations, and elucidates the impacts of simulated OH concentration fields on CH4 using the LMDz chemical transport model. The manuscript contains novel investigation to reveal inter-annual variations in OH and its impact on CH4 over recent decades using multi-model approach. The topic of the manuscript is certainly within the scope of ACP. Overall, the manuscript is well written and easy to follow. I would like to consider the publication of the manuscript from ACP, while I have several comments below which should be addressed before publication.

**Response:**
**We thank the reviewer for the helpful comments. All of them have been addressed in the revised manuscript. Please see our itemized responses below.**

**Specific comments:**
**Comments:** OH field Is the prescribed biogenic NMVOC emissions (p. 8, l. 187) climatology? Please clarify.
**We mean here that some models just prescribed a fix scenario for NMVOC emissions and do not accoiunt for time variability.**
**Text has been clarified : "Biogenic NMVOC emissions in CESM and GEOSCCM are calculated based on the distribution of plant functional types and meteorology conditions with MEGAN, whereas the other models prescribe climatological biogenic NMVOC emissions."**

**Comments:** How did the authors prescribe the ECLIPSE and RCP85 emission inventories in the INCA simulations during the periods before 2004, between 2006-

2009, and after 2011?

**Response: We clarify by changing this sentence to" Anthropogenic emissions from Short-Lived Pollutants (ECLIPSE) inventory (Stohl et al., 2015) for 2005 and RCP 85 emission inventory (Riahi et al., 2011)) for 2010 are applied to every year of INCA NMHC-AER-S and INCA NMHC simulations, respectively."**

**Comments:** 2.2.2. Model simulations Please clarify how the OH increasing and decreasing rates are determined in the Run_OH_inc and Run_OH_dec simulations. Why are the rates +1 and −1.

**Response: We clarify by add in the text : " In order to assess the recent change in [OH], we tested two additional scenarios between 2010 and 2016: one with [OH] increase of $+0.1\%$ $yr^{-1}$ (Run_OH_inc) according to the slightly changing of OH calculated by ACCMIP models and one with [OH] decrease of $-1\%$ $yr^{-1}$ (Run_OH_dec) according to obviously decreasing of OH calculated by top-down approaches constrained by observations."**

Comments: 3.1. Spatial distributions of tropospheric OH The authors attributed possible causes of too large interhemispheric differences in OH in the CCMI models to model O3 and CO biases and unaccounted processes in some of the CCMI models, as reported by previous studies. Why is not the model performance on O3 and CO in the CCMI ensembles evaluated or referred? It might be better to cite Strode et al. (2016), Revell et al. (2018), and other papers.

**Response: We acknowledge that the depth of analysis of the root causes of what we find here can be increased the lack of evaluation of these models in our paper. We have added in the text:**
**"Previous studies have attributed the inconsistency between the simulated and the observed OH N/S ratios to a model overestimation of O₃ and underestimation of**

**CO over the Northern Hemisphere (Naik et al., 2013; Young et al., 2013; Strode et al., 2015), which have also been reported for CCMI models (Strode et al., 2016; Revell et al., 2018), …"**

**Add references:" Strode SA et al. (2016) Interpreting space-based trends in carbon monoxide with multiple models Atmos Chem Phys 16:7285-7294 doi:10.5194/acp-16-7285-2016"**

**We have increased the depth of the analysis of the root causes possibly explaining what we find in the paper, all along with the text (see answers to reviewer 1).**

Comments: 3.3. Factors contributing to inter-model differences Why the authors did not assess inter-model differences in tropospheric O3 burden? The tropospheric O3 burden should also affect primary production of OH.

**Response: We have calculated global mean O3 mixing ratios averaged over the tropospheric and their pressure altitude levels in table 5 and move the values of O1(D) and reactive humidity, which contribute less to the inter-model difference of [OH] to the supplement (Table S4).**

| | CO ppbv | | | | NO pptv | | | | O3 ppbv | | | |
|---|---|---|---|---|---|---|---|---|---|---|---|---|
| | 750 | 500 | 250 | Tp | 750 | 500 | 250 | Tp | 750 | 500 | 250 | Tp |
| **CESM1-CAM4Chem** | 76 | 71 | 70 | 71 | 9 | 4 | 12 | 13 | 32 | 42 | 57 | 48 |
| **CESM1-WACCM** | 75 | 70 | 69 | 70 | 9 | 5 | 12 | 12 | 31 | 41 | 55 | 47 |
| **CMAM** | 77 | 68 | 64 | 69 | 17 | 4 | 17 | 26 | 34 | 43 | 60 | 52 |
| **EMAC-L47MA** | 85 | 77 | 70 | 75 | 8 | 4 | 11 | 14 | 38 | 48 | 63 | 56 |
| **EMAC-L90MA** | 84 | 76 | 69 | 74 | 8 | 5 | 11 | 17 | 38 | 48 | 61 | 54 |
| **GEOSCCM** | 78 | 74 | 73 | 74 | 9 | 5 | 13 | 13 | 33 | 43 | 61 | 49 |
| **MOCAGE** | 67 | 68 | 67 | 67 | 26 | 14 | 17 | 20 | 37 | 42 | 46 | 43 |
| **MRI-ESM1r1** | 93 | 86 | 83 | 86 | 10 | 5 | 20 | 32 | 36 | 48 | 67 | 56 |
| **SOCOL3** | 79 | 73 | 74 | 74 | 48 | 10 | 14 | 25 | 43 | 54 | 67 | 61 |
| **Mean ±stand. dev.** | 79±7 | 74±6 | 71±5 | 73±5 | 16±13 | 6±3 | 14±3 | 19±7 | 36±4 | 45±5 | 60±7 | 52±6 |

**And we have added in the text:**

**" To analyze inter-model differences in OH vertical distributions, we compared CO, NO, and O$_3$ mixing ratios in table 5 as well as O($^1$D) photolysis rates and**

**specific humidity in Table S4.**

**"Tropospheric $O_3$ can also influence primary production of OH, and tropospheric $O_3$ burden reflects combined effects of $NO_x$, CO, and VOCs. The high $O_3$ over the lower troposphere simulated by SOCOL3 and the low $O_3$ over the upper troposphere simulated by MOCAGE can contribute to explain the high and low [OH] simulated the two models over the corresponding altitudes, respectively. "**

**Comments:** Do inter-model differences in vertical distribution of lighting NO production affect OH vertical distributions?

**Response:**

**Yes indeed. We have added table S3 in the supplement:**

**Table S3. Lighting NOx emission (Tg N $yr^{-1}$) over three pressure altitudinal intervals and the total troposphere of CCMI models over 2000-2010.**

|              | Surface-750hPa | 750-500hPa | 500-250hPa | 250-100hPa | tp   |
|--------------|----------------|------------|------------|------------|------|
| CMAM         | 0.7            | 0.4        | 1.5        | 1.7        | 4.2  |
| EMAC-L90MA   | 0.2            | 0.5        | 1.3        | 1.8        | 3.7  |
| CESM1-WACCM  | 0.2            | 0.6        | 2.7        | 0.7        | 4.2  |
| GEOSCCM      | 0.2            | 1.3        | 3.3        | 0.8        | 5.6  |
| MOCAGE       | 0.3            | 1.2        | 2.4        | 1.0        | 4.8  |
| MRI-ESM1r1   | 1.4            | 0.7        | 3.2        | 5.2        | 10.2 |
| SOCOL3       | 0.2            | 0.8        | 2.1        | 1.4        | 4.4  |

**We have also added in the text:"**

**L431-L435: "Lighting $NO_x$ emissions, which are mainly emitted in the middle and upper troposphere, can contribute to inter-model differences in NO and OH distributions (Murray et al., 2013; 2014). We compare lighting $NO_x$ emissions calculated by CCMI models in Table S3.. High lighting NOx emissions simulated by MRI-ESM1r1 above 250hPa can explain high NO mixing ratios and increasing OH with altitude over the upper troposphere for this model (Fig. 3). However, High NO in the lower troposphere simulated by MOCAGE and SOCOL3 are not corresponding to high lighting $NO_x$ emissions in these models."**

**L451: Lighting NOx emissions range from 3.7-10.2 Tg yr$^{-1}$(table S3)**

**Comments:"** 3.4. Inter-annual variations of OH What is possible cause of significant positive [OH] trends over the tropics (p. 19, l. 454)?**"**

**Response: We add in the text:" By comparing spatial distribution of OH trend with specific humidity (Fig.S6a), NO$_x$ and CO emissions (Fig. S6b), and stratospheric O$_3$ (Fig.S6c), we find that positive OH trend over tropical regions are mainly corresponding to increases in water vapor (Fig. S6a)"**

**And we add figure S6a in the supplement:**

[Figure]

**Figure S6a.** Spatial distribution of tropospheric specific humidity trends from 2000 to 2010 (in 10$^{-2}$ g/kg year$^{-1}$). Black dots denote model grid-cells with statistically significant trends (p-value < 0.05).

**Comments:** 4.2.1. Spatial distributions of tropospheric CH$_4$ mixing ratio Could you explain how inter-model differences in spatial and temporal OH variations affect the simulated global CH4 mixing ratio more in depth?

**Response: In our paper, we attribute differences in LMDz simulated global mean CH4 mixing ratio to different global OH mean value and trend, and the spatial distribution of CH4 to multi-model spread in OH spatial and temporal distributions. To clarify our point, we have re-organized the first paragraph of section 4.2.1:**

**" We used the scaled OH fields to perform simulations between 2000 and 2010. Figure 6 shows the spatial distribution of tropospheric $CH_4$ mixing ratios for the simulation Run_standard (Table. 2, driven by OH with inter-annual variations) averaged over 2000-2010. Although all simulations started from the same initial conditions and OH fields were scaled to give the same global $CH_4$ loss as INCA NMHC in 2000, LMDz simulations using the different scaled OH fields still generated a spread of tropospheric mean (8 ppbv) and spatial distribution in $CH_4$ mixing ratios averaged during 2000-2010. Differences between the global tropospheric mean [OH] cannot explain these differences (see Table 4). Clearly, the different spatial (horizontal and vertical) and temporal variations of the OH fields (as described in Sect. 3), which were kept in this experiment by only scaling [OH] globally , significantly modify the simulated $CH_4$ mixing ratios (Table 7 and Fig. 6). OH fields with increasing trend will lead to lower LMDz simulated $CH_4$ mixing ratios. The LMDz simulation using the TransCom OH fields (without inter-annual variability) shows the highest $CH_4$ mixing ratios (1735 ppbv), while the one using the CMAM OH (with slightly increasing OH trend during the decade) shows the lowest $CH_4$ mixing ratios (1727 ppbv). "**

**And we add in the second paragraph:" The differences in spatial distribution of OH fields can influence LMDz simulated $CH_4$ spatial distributions."**

**Comments:** Technical corrections: p. 13, l. 317: publication year is missing.

**Response: We add the publication year, thank you very much for pointing out.**

[revised manuscript text omitted]